# Bluetooth-Based Healthcare Information and Medical Resource Management System

**DOI:** 10.3390/s23125389

**Published:** 2023-06-07

**Authors:** Chao-Shu Chang, Tin-Hao Wu, Yu-Chi Wu, Chin-Chuan Han

**Affiliations:** 1Department of Information Management, National United University, Miaoli 36003, Taiwan; 2Department of Electrical Engineering, National United University, Miaoli 36003, Taiwan; 3Department of Computer Science and Information Engineering, National United University, Miaoli 36003, Taiwan

**Keywords:** Internet of Things, wearable devices, physiological information, healthcare information, medical resource management, MQTT, indoor positioning

## Abstract

This paper presents a healthcare information and medical resource management platform utilizing wearable devices, physiological sensors, and an indoor positioning system (IPS). This platform provides medical healthcare information management based on the physiological information collected by wearable devices and Bluetooth data collectors. The Internet of Things (IoT) is constructed for this medical care purpose. The collected data are classified and used to monitor the status of patients in real time with a Secure MQTT mechanism. The measured physiological signals are also used for developing an IPS. When the patient is out of the safety zone, the IPS will send an alert message instantly by pushing the server to remind the caretaker, easing the caretaker’s burden and offering extra protection for the patient. The presented system also provides medical resource management with the help of IPS. The medical equipment and devices can be tracked by IPS to tackle some equipment rental problems, such as lost and found. A platform for the medical staff work coordination information exchange and transmission is also developed to expedite the maintenance of medical equipment, providing the shared medical information to healthcare and management staff in a timely and transparent manner. The presented system in this paper will finally reduce the loading of medical staff during the COVID-19 pandemic period.

## 1. Introduction

The development of the Internet of Things (IoT) [1,2,3] has led to the rise of wearable mobile devices, which can be used to record physiological data, such as heart rate and exercise status via smartwatches, or to display information on wearable optical glasses in a hands-free manner, such as Google Glass. Wearable mobile devices have been applied in various fields, especially in the field of medical care. Wearable mobile devices can be used to help measure and record patients’ physiological data or be installed on medical devices to record information and usage records of medical devices to help healthcare professionals manage them [4,5,6]. Medical care and healthcare are thus hot areas of IoT applications [7]. Through the technology of IoT, various medical sensor devices and wearable devices are integrated, and the information from each sensor device is received through a wireless network [8,9,10], improving the service quality and efficiency in the field of medical care and healthcare [11] as well as helping medical institutions to manage healthcare information and medical resources [12].

Advances in medical care have led to a significant reduction in the death rate and a gradual increase in average life expectancy. According to the Ministry of the Interior, Taiwan, the elderly population exceeded 14% of the total population in 2018, meaning Taiwan officially became an “aging society”. As the proportion of the elderly population continues to rise, the demand for medical care escalates, and the workload of healthcare workers increases. This, coupled with the declining fertility rate, has resulted in a declining workforce and has highlighted the shortage of medical manpower. In addition to patient care, medical staff in medical institutions also need to manage medical equipment and supplies [13]. Medical devices and equipment such as wheelchairs and crutches are available for rent to people/patients who come to the facility, but people often forget to return them to their original place after using them and may even take them away from the facility. These oversights often cause great problems for medical staff in the management of medical supplies and increase the workload of medical staff. Furthermore, in the maintenance of medical equipment, management staff need to be quickly informed of the maintenance time and status of each piece of medical equipment. For example, in the case of bed management, the statuses of beds, such as not-in-use, in-use, and clearing beds, are required to be updated immediately so that medical staff can clearly have those empty beds for new patients. During the period between patient check-out and bed cleaning, it is often necessary for nursing staff and bed cleaning staff to provide each other with real-time information on the status of beds to improve the overall speed of the bed cleaning process. In the past practice of using telephone contact for bed cleaning, nursing staff may forget, due to their busy work schedule, to notify bed cleaners in time when patients check out, which leads to delays in bed cleaning progress. Therefore, how to provide this shared medical information to healthcare and medical management staff in a timely and transparent manner to enhance the efficiency of medical institutions is an urgent problem for medical institutions today.

Therefore, in order to provide a solution to the above challenges, this paper will apply wearable devices in medical institutions and physiological information sensing technology to build a medical Internet of Things (IoT) platform and integrate various medical-related and healthcare-related information through the MQTT (Message Queuing Telemetry Transport) protocol of IoT [14,15,16,17]. An information security protection method during the information transmission process [11] is also proposed to tackle the shortcomings of the MQTT (Message Queuing Telemetry Transport) protocol in information security protection. For medical information management, the physiological information of patients measured by various medical sensors is collected, classified, integrated, and transmitted to a database [18]. Approaches to integrating the physiological information collected by the original equipment of medical institutions are proposed. For medical resources management, the hospital equipment/devices are located and tracked by the developed positioning technology [19] to solve the problem of equipment loss that occurs when medical equipment is rented. Therefore, indoor positioning for these medical devices and equipment can reduce the inconvenience of device management and improve management efficiency. This indoor positioning tracking can also be used to give warnings for further safety management of patients or care recipients. For example, when a patient or care recipient enters a dangerous area of the hospital, such as a stairway area, or when a care recipient is found to be leaving the hospital through the entrance area with the driveway outside, the positioning information in these areas provides warnings to keep the patient or care recipient from being injured.

The presented system establishes an “information exchange platform for medical personnel”, through which medical personnel not only can release official relevant information but also can obtain all the medical information mentioned above so that medical personnel can keep track of the patient’s physical condition and the location [20,21,22,23] and the borrowing status of medical equipment/devices. Through this study, it is expected to reduce the management burden of medical staff and accelerate the management efficiency of medical staff. The main contributions of this paper are summarized as follows.

(1)This paper integrates physiological information sensing technology and wearable devices with mobile positioning technology in an Internet of Things-based medical care and medical resource management platform.(2)This IoT platform uses wearable devices and physiological information sensing technology to perform real-time patient physiological information monitoring and medical care information management and uses a proposed indoor positioning tracking to establish a safe activity area for the care recipient, allowing the caregiver to understand the patient’s physiological information and movement, assisting and reducing the caregiver’s burden and providing an additional layer of a protection mechanism for the care recipient.(3)The IoT platform combines indoor positioning tracking for hospital equipment and device management, solving the problem of medical equipment rental management.(4)The IoT platform provides a real-time information exchange function, allowing medical staff and medical management staff to share medical information to improve the efficiency of medical institutions.(5)This system reduces the workload of nursing and medical management staff and improves the efficiency and quality of care with limited human resources.

## 2. Related Work and Literature Review

### 2.1. Remote Health Monitoring Systems

Remote Health Monitoring Systems (RHMS) [24] based on Wireless Body Area Networks reduce the number of unessential hospitalizations by providing long-term management of health conditions, disease prevention, and detection of emergencies for patients at home. Furthermore, continuous health monitoring using RHMS is a hopeful solution for elderly people suffering from chronic diseases. RHMS is based on the deployment of a Wireless Body Area Network (WBAN) [25], called WBAN-based RHMS (WBAN-RHMS), using wearable and/or implantable sensors in or around the human body. The sensed physiological data are forwarded through a wireless Bluetooth network to a collecting node known as a data collector or a gateway or coordinator (smartphone or PDA: Personal Data Assistant), which is connected to the remote server (for data processing and storage) through the Internet network (via WiFi) or the cellular networks (4G or 5G) as Figure 1.

#### 2.1.1. Tier 1: A Sensor Node

Figure 1 shows a four-tier architecture of a Remote Health Monitoring System based on a Wireless Body Area Network (WBAN-RHMS) [26]. Tier 1 is about the sensor node, which senses physiological data of the human body. The sensor node is the source of information from either a patient with a particular disease or an athlete on a workout, and the sensed data could be forwarded to the BAN coordinator through WiFi or Bluetooth transmission.

#### 2.1.2. Tier 2: The BAN Coordinator

The body area network coordinator (BAN coordinator), which can be a smartphone, a PDA, or a laptop, collects sensed data from implantable and/or wearable sensors and uploads them to the remote station via one of the available networks (WIFI, 4G/5G, etc.) for storage and future processing.

#### 2.1.3. Tier 3: Data Storage and Processing

The data storage and processing part of the system serves as a database and can be a cloud, a dedicated server, a web server, or a local server.

#### 2.1.4. Tier 4: Monitor

This part is related to the monitor that represents the person who looks after changes in the patient’s life parameters. This monitor module obtains access to the data stored in a remote database for viewing or analyzing or else editing some related recommendations.

The basic concepts related to WBAN-RHMS design and development were introduced in [24]. Besides that, the exploration of the BLE communication protocol was also discussed. Particularly, the investigation of the first tier is mainly addressed, and the selected communication technology between the sensor (Tier 1) and the coordinator (Tier 2) is Bluetooth Low Energy (BLE). Therefore, the first tier, which is composed of wearable sensors that sense the body’s vital signs (e.g., body temperature, heart rate, SpO2, ECG, EEG, etc.) through the sensor nodes and sends them to the smartphone via BLE communication interfaces, was illustrated in detail. However, it only focuses on the description of wearable sensor devices of Tier 1 and BLE interface connection, but the data transmission methods between Tier 2 and Tier 3, as well as the integration and application of various sensor sensing data, are not explained in detail. Moreover, the monitoring method of Tier 4 is also not explained.

### 2.2. Internet of Things and Medical Management

The term Internet of Things (IoT) was first proposed by Kevin Ashton in a speech in 1999 when the communication technology only included RFID communication technology, and the ITU (International Telecommunication Union) report on the Internet of Things [1] mentioned that the Internet of Things technology should not be limited to RFID, but should also be able to apply other communication technologies as the basis of the Internet of Things. The initial definition of the Internet of Things is that all objects in IoT must be defined by a standard protocol to define their global network address. Since then, IoT has started to flourish in various fields, among which the medical care and healthcare field is a very popular field. In this field, the main purpose is to enhance the service quality and efficiency of medical care and healthcare through the application of IoT technology and to accelerate the development of knowledge [7,27].

Nowadays, the most frequently proposed application in the management of medical institutions is to connect medical devices through IoT technology to collect various physiological data. For example, Zhang et al. [9] proposed the use of IoT technology to observe patients’ intravenous injections. Satija et al. [8] and Nurdin et al. [4] used IoT to monitor patients’ ECG and send this information to physicians for condition analysis in real time. These researches are all conducted through IoT technology to collect the physiological data of hospital patients and send the information to healthcare workers so that healthcare workers can monitor the real-time condition of patients in charge at any time and anywhere. In addition, IoT technology can also be applied to equipment resource management; for example, Xu et al. [12] proposed the use of IoT technology to manage the number and location of emergency medical equipment so that the relevant equipment can be prepared as quickly as possible in case of medical emergencies. Farhat et al. [28] proposed installing sensing devices on medical equipment, observing the operation of the equipment through IoT technology, and predicting the remaining life of the equipment by the collected data.

At present, Internet of Things technology has been gradually applied in medical institutions, as it is the collection of physiological information or the management of resources, and it can greatly reduce the burden of medical staff and alleviate their work pressure. However, in medical institutions, with the diversification of IoT application technologies, such as the collection format of patient physiological information and the positioning information of medical equipment, the data format of this information may be different; therefore, healthcare workers may need to constantly switch between different platforms for information exchange and use. Therefore, this study intends to integrate various information formats in medical institutions into one platform so that healthcare workers can be more consistent and convenient in integrating and exchanging information.

### 2.3. Message Queuing Telemetry Transport Protocol

Message Queuing Telemetry Transport (MQTT) Protocol is an important protocol that is more commonly used in IoT. It is based on TCP/IP connection and can transmit and receive messages with light network bandwidth and computational load in restricted environments such as low bandwidth and unstable network connections.

MQTT’s message delivery uses the Publish/Subscribe mode, which contains four important elements: Publisher, Subscriber, Topic, and Broker, and provides three QoS levels for message delivery services for users to choose from. The data flow of transmission is shown in Figure 2. The Broker is a message relay station. It transfers messages within a Topic from a Publisher to a Subscriber. A Topic is a keyword used by the MQTT Broker to filter messages against the MQTT client. Users in Subscriber mode inform the message relay (Broker) of the Topic they want to register and become the Subscriber of this Topic. When messages from Publishers were sent to this Topic, the relative Subscriber would obtain these messages. Each Subscriber can register for more than one Topic.

The Publisher represents the user who sent the message. When the Publisher sends a message, it is not sent directly to the Subscriber, but the message is sent to the message relay (Broker), and the message relay (Broker) is told to send the message to the specified Topic, and the message relay (Broker) will re-send the message to all the Subscribers who have registered for the specified Topic. The MQTT protocol provides three QoS (Quality of Service) Levels for users to choose from during message transmission.

#### 2.3.1. QoS Level 0: At Most Once

In the case of QoS Level 0, the message is distributed from the Publisher to the message relay (Broker), and the message relay (Broker) then distributes the message to all Subscribers. At this QoS Level, the MQTT protocol only ensures that messages are sent at most once but does not guarantee that registrants actually receive the messages. Therefore, with a QoS Level of 0, messages may be lost during transmission due to network outages and other factors, resulting in messages not being received by the message Broker or Subscriber and lacking reliability in data delivery. Therefore, this mechanism is more suitable for the transmission of unreliable sensor messages.

#### 2.3.2. QoS Level 1: At Least Once

In the case that QoS Level is set to 1, this mechanism ensures that the target receiver will receive the message at least once after the message is published. The Publisher sends the message to the message relay (Broker), and the message relay (Broker) will reply with a PUBACK (publish acknowledgement) message to inform the Publisher that the message has been received by the message relay (Broker). The same process is used to send messages from the message relay to the Subscriber, and the Subscriber will reply to the PUBACK message to the message relay after receiving the message.

When the PUBACK message is not received from the receiver within a certain period of time after the message is sent, the sender will consider that the message has not been sent and start to try to re-send the message again and again until the PUBACK message is received. However, when the PUBACK message is lost in the transmission process due to the network factor after the receiver replies to the PUBACK message, and the Publisher also sends the message again because the PUBACK message is not received, the situation that the receiver receives repeated messages will occur.

#### 2.3.3. QoS Level 2: Exactly Once

With the QoS Level set to 2, this mechanism ensures that the receiver will definitely receive the message and that the receiver will only receive the message once, as shown in Figure 3. After the Publisher has sent the message to the message relay (Broker), the message relay (Broker) will reply with a PUBREC (Publish Received) message to the Publisher, informing the Publisher that the Broker has actually received the published message and the Publisher will send a PUBREL (Publish Release) message after receiving the PUBREC message. After receiving the PUBREL message, the message relay station (Broker) will first publish the message to the Subscriber and then return the PUBCOMP (Publish Complete) message to the Publisher. After receiving the PUBREL message, the message will be sent to the Subscriber, and then PUBCOMP (Publish Complete) will be sent back to the Publisher. This mechanism uses PUBREC and PUBREL messages to ensure that the receiver will always receive the message and will not receive duplicate messages.

### 2.4. Related Research

Adbulmalek et al. [29] provided a systematic literature review on recent studies of IoT-based healthcare-monitoring systems, compared various systems’ effectiveness, efficiency, data protection, privacy, security, and monitoring, and discussed the challenges and open issues regarding security and privacy and Quality of Service (QoS). Heaney et al. [30] presented a custom-built, end-to-end ECG (Electrocardiography) capturing prototype system. This system utilizes an AD8232 microchip (Analog Devices, Wilmington, MA, USA) as the analog front-end, which is then fed into an Arduino (MKR1010, Arduino, Italy), which offers both Bluetooth and WiFi connectivity to a smartphone or another external device for the remote monitoring of the patients. The system is also equipped with a temperature and specific oxygen (SpO2) sensor and is able to display an ECG signal/SpO2/temperature on a local display screen, on a custom mobile application through Bluetooth, and on a server through WiFi.

Tang et al. [31] proposed a 5G-based architecture for smart healthcare information infrastructure, which optimized the latest technical architecture standardized by 3GPP (3rd Generation Partnership Project) and ETSI (European Telecommunications Standards Institute) about MEC (multi-access edge computing) 5G Integration under a smart healthcare scenario. A new network element, iGW (industry gateway), as the core was defined, and a smart healthcare dedicated cloud platform, iMEP (industry multi-access edge platform), was also introduced. Different types of physiological signals were acquired using GSR (Galvanic Skin Reaction), EMG (Electromyography), and HRM (Heart Rate Monitor) sensors and were analyzed in terms of variability over time before, during, and after the EMDR (Eye Movement Desensitization and Reprocessing) session [32], in which a VR (Virtual Reality) with the bilateral stimulation used in as a tool to relieve stress was proposed. A 15 min relaxation training program was created for adults in a virtual, relaxing environment. The data from the sensors placed in or on the patient’s body contain personal private information, and their transmission out of the body towards a medical server, therefore, requires communication security.

Azbeg et al. [33] presented a secure healthcare system that integrates IoT with Blockchain to support remote patient monitoring, especially when it comes to chronic diseases that need regular monitoring. The security was ensured by using the re-encryption proxy together with Blockchain to store hash data via IPFS (Inter Planetary File System). Mucchi et al. [34] reviewed the security challenges and solutions for WBAN (Wireless Body Area Network). The challenges include data confidentiality, data authentication, data integrity, data freshness, secure management, data availability, efficiency and usability, authorization, non-repudiation, scalability up and downsizing, flexibility, data security, and reliability. Some security strategies that satisfy the requirements for securing internal communication in WBAN are avoiding DoS (denial-of-service) attacks, avoiding data tampering attacks, achieving data confidentiality, and avoiding data authenticity. Discretionary Access Control (DAC), Mandatory Access Control, Role-Based Access Control (RBBAC), and Attribute-Based Encryption (ABE) are some of the existing access control adopted algorithms. A trade-off among security, efficiency, flexibility, and usability should be made for new solutions to cope with the future trend of the development of smaller and low-power consumption medical sensors.

## 3. Materials and Methods

In today’s environment of a severe shortage of medical manpower, the problem of high nurse–patient ratio, which means that each healthcare worker is often responsible for the care of several patients, puts the healthcare workers in medical institutions under considerable work pressure [35]. In addition to patient care, the management of medical equipment and devices is also part of the work of healthcare workers. However, due to the large number and complexity of medical equipment and devices, management problems can easily arise [28]. For example, many devices are often not returned after being rented by patients or other healthcare workers, and the incorrect quantity of these devices is only found when taking inventory of the equipment, requiring time to retrieve these devices and thus creating an additional workload for healthcare workers.

In this study, we developed a Bluetooth data collector, an App, and a cloud server to build an Internet of Things (IoT) environment for healthcare information/medical resource management in an integrated manner based on the temperature measurement smart T-shirt (body temperature sensor) and the smart adult diaper (diaper humidity sensor) developed by Sinopulsar [36] as shown in Figure 4 and Figure 5. The MQTT protocol of the IoT is used to integrate various medical and healthcare-related information. First of all, in healthcare information management, we developed a Bluetooth data collector to steadily collect, classify, and integrate patient physiological information measured by various medical sensors and then transmit it to a database for storage. Secondly, in medical resource management, we managed to locate and track hospital equipment and devices through the deployment of positioning devices. Finally, the system establishes a dissemination server responsible for the timely dissemination of information in response to alert messages.

Therefore, this study implements a medical information management platform through the Internet of Things (IoT) transmission protocol, MQTT, to integrate the work information of healthcare workers, and its protocol hierarchy is shown in Figure 6. The platform provides three different types of agents, namely, Monitoring Agent, Positioning Agent, and Exchanging Agent. Medical Data Monitoring, Medical Equipment Positioning, and Message Exchanging help healthcare professionals quickly grasp the real-time situation of patients and track the positioning of medical equipment and devices to facilitate the management of medical staff, reducing the workload of medical staff. In addition, the presented system also copes with the lack of security mechanism in the information transmission process of the MQTT protocol and conducts data content protection [37], which is to design a security mechanism (Secure MQTT) on top of the MQTT protocol to encrypt and protect the content of the information transmission. The platform allows healthcare users to transfer and exchange data with confidence to avoid the leakage of patients’ private information.

Figure 7a shows the system usage scenarios, where care recipients wear smart T-shirts/diapers in the ward, and the Bluetooth Hub at the entrance of the ward and the APP-Hub (either BT or WiFi) on the smartphone are running the information collection program to collect the body temperature and diaper humidity information of the care recipient through the multi-channel Bluetooth transmission protocol. Figure 7b,c illustrate the workflow chart for smart sensors used to collect physiological data and Bluetooth tags used for indoor positioning, respectively. The Bluetooth Hub sends physiological data to the system server. The system server can transmit notifications/alerts (work coordination information exchange or indoor position information) to smartphones via WiFi/4G. Figure 8 shows the screen of the APP-Hub program to collect temperature/humidity information. The screen shows that the body temperature and diaper humidity information from different care recipients is collected, and the information is then forwarded to the back-end physiological information collection server for monitoring via WiFi/4G wireless transmission technology. Figure 9 shows the pairing screen of the patients and their physiological sensors.

Figure 10 shows the humidity information screen of the back-end physiological information collection server. The physiological information monitoring module then monitors the humidity information, and when the humidity information increases to a certain threshold value (e.g., humidity information exceeds 70%), the alert server will request the alert message push server to notify the caregiver of the patient.

The architecture of the presented system is shown in Figure 11, which is divided into System Server and Service Module. The MQTT Broker server is installed in the back-end server to receive the information collected in the medical institution, and the medical information is mainly from the following three sources:Patient physiological monitoring information;Location information of medical personnel and equipment;Medical staff work coordination information (message).

The MQTT Broker server at the back-end of the system receives the above information and sends the processed data to the back-end database through the Data Processing Module using the HTTP transmission protocol, while users can also retrieve the content of the back-end database through the system platform.

### 3.1. System Server

#### 3.1.1. MQTT Broker

In the MQTT protocol, all messages are first sent to a message Broker, which forwards messages to a Subscriber based on a Topic, and Eclipse’s Mosquitto server is used as the message Broker for the presented system platform. Mosquitto Server is an open source MQTT server developed for the Eclipse Foundation’s Open Source IoT project and can be installed and run on Windows, Linux, and Mac OS operating systems to run the MQTT protocol.

To facilitate the management of Topics on this platform, a hierarchical approach to Topic naming is used. The three sources of medical information are used as the first level of Topics, as shown in Table 1. Under the hierarchical naming approach, the slash symbol “/” is used to categorize the Topics of different departments and the hierarchical structure of naming is shown in Figure 12. In Figure 12, we can see that the Topics are divided into three levels based on the departments. According to this hierarchy, the theme names of the department are shown in Table 2.

#### 3.1.2. Data Processing Module

A data processing module is set up at the back-end of the system. This module interfaces with the MQTT Broker and is responsible for the following tasks:(a)Data packet format parsing: The data processing module will register all the Topics on this platform, so this module will receive all the messages transmitted on the Topics, parse the JSON format of the messages after receiving them, and save the data to the database after obtaining the packet contents.(b)Generating a “system symmetric key”: To protect the security of message transmission on this platform, the data processing module generates a “system symmetric key” to encrypt and decrypt messages transmitted on the platform. The data processing module will also send this symmetric key to all clients.(c)Equipment positioning information calculation: In the location tracking [2] of medical personnel and equipment, the located equipment will put the IDs of the three closest Bluetooth Beacon devices, i.e., RSSI signal strength values, into the medical information (Position Information) packet and send it back. After parsing the packet, the data processing module will obtain the coordinates of the Beacon devices from the Beacon Table based on the returned Beacon device IDs and perform the indoor positioning algorithm with the RSSI signal strength values to calculate the current coordinates of the positioned devices. Finally, the coordinates are updated to the Equipment Table in the database.

#### 3.1.3. Database

A MySQL database will be set up at the back-end of the system to store sensor measurement data and medical device positioning information. The data tables are divided into (1) User Data, (2) Measurement Data, (3) Position Information, and (4) Message Data, and the schematic diagram of the data table is shown in Figure 13.

#### 3.1.4. Data Transmission Format

In order to integrate the content format of data messages, JSON is used as the main data transmission format. The types of data to be transmitted are sensor data, device location information, and message exchange data. The sensor data include the data type, address of the sensor entity, and data (Figure 14a); the device location information includes the address and distance of the device entity (Figure 14b); and the message exchange data consist of the name of the transmission topic and the message content (Figure 14c). Take the device positioning information in Figure 14b as an example; the Type field is used to define the type of data in this JSON format. “Positioning” means that this data is related to positioning, and the object of “Data” packs the data we want to transmit. The content of the packed data in Figure 14b includes the MAC of the device (the value of “Address” is 8EAD568DE61A), the location of the device, etc.

### 3.2. Service Module

#### 3.2.1. Patient Physiological Monitoring Information

Patient physiological data measured by medical sensors can be transmitted through the MQTT protocol to the healthcare provider in charge of the patient so that the healthcare provider can monitor and manage the patient’s condition in real time at any time.

In this study, an Arduino microcontroller and a DHT11 temperature/humidity sensor were used to simulate a medical sensor device to send the patient’s body temperature/humidity information. The actual temperature/humidity data of the patients are measured by the smart T-shirt and smart adult diaper. Through the Publish/Subscribe transmission mode of the MQTT protocol, a dedicated physiological monitoring information topic (Topic) is created, and the sensed physiological information (e.g., body temperature or diaper humidity) is published to this Topic. At the same time, the healthcare personnel responsible for the patient’s care can register for this Topic so that the healthcare personnel who have registered to this Topic can continuously receive the latest physiological conditions of the patient to confirm the effectiveness and correctness of the MQTT protocol.

The temperature measurement smart T-shirt and the smart adult diaper [36] are then applied to the system shown in Figure 7 to transmit the actual physiological information. The Bluetooth Hub at the entrance of the ward or the APP-Hub (BT/WiFi) on the smartphone will run a temperature and diaper humidity information collection program to collect the temperature and diaper humidity information of the care recipient using the multi-channel Bluetooth transmission protocol. The smartphone or Bluetooth Hub will then transmit the collected physiological information to the back-end database for storage and monitoring (Figure 10) using the MQTT protocol.

When the information monitoring server finds that the monitored information is abnormal or in an emergency situation, the monitoring server will then push the abnormal value and the emergency situation to the relevant healthcare and management personnel through the push server for emergency disposal. The abnormal values are physician-defined measurement criteria. The data transmission process is shown in Figure 15.

In addition to collecting data from devices through cell phones in Bluetooth monitoring mode, the data collector (Hub) will use Bluetooth to collect data from all devices in the range via broadcasting and upload them to the back-end System Server/Data Server for storage. When there is an abnormality in the data, the information monitoring server sends a notification to the Push Server and then pushes the notification to the caregiver’s cell phone to remind the caregiver of the abnormal condition, such as the patient’s body temperature is over 38 or below 32 degrees, the diaper humidity is higher than 85%, or even the sensor tag power is below 20%, etc.

#### 3.2.2. Positioning Information of Medical Personnel and Equipment

This type of information is mainly generated from the location tracking of medical devices, equipment, or medical personnel. In addition to collecting physiological information, Bluetooth broadcasting signals from Bluetooth beacons placed on medical devices and equipment can also be collected through the Bluetooth Hub equipment deployed in medical institutions [38,39,40]. This will reduce the inconvenience of device management and improve management efficiency. In addition to the tracking of medical devices and equipment, the Bluetooth Beacon can also be used to track the location of medical personnel.

The Bluetooth Hub device is widely deployed in the medical facility, and the area of the medical facility is coordinatized (red dots as shown in Figure 16), and the location is tracked by the coordinates of the Bluetooth Hub device and the RSSI signal strength value in the Beacon broadcast packet. A positioning algorithm based on the circle–circle intersection points was developed. The RSSI signal strength value is obtained from the Bluetooth Beacon packets collected from medical personnel or medical devices. This value is converted into the distance and used as the radius of a circle, and the coordinates of the medical device (point G in Figure 17) are calculated based on the intersection of the circles, as shown in Figure 17. In this study, instead of finding the intersection points of two circles, a different calculation method is used to find the centroids of these two intersection points. Circle *A* and circle *B* are first taken out (as in Figure 18), and according to the Pythagorean theorem, the following Equations (1)–(3) can be derived, where $\bar{AB}$ represents the distance between *A* and *B*. Line $\bar{QP}$ is perpendicular to Line $\bar{AB}$ and the intersection point is point *E*; therefore, Equations (2) and (3) hold.
(1)$$\bar{AB}=\bar{BE}+\bar{AE}$$
(2)$${\bar{AQ}}^{2}={\bar{AE}}^{2}+{\bar{QE}}^{2}$$
(3)$${\bar{BQ}}^{2}={\bar{BE}}^{2}+{\bar{QE}}^{2}$$

Subtract Equation (2) from Equation (3) and use Equation (1) to solve for $\bar{BE}$, yielding
(4)$${\bar{BQ}}^{2}-{\bar{AQ}}^{2}={\bar{BE}}^{2}-{\bar{AE}}^{2}={\bar{BE}}^{2}-{\left(\bar{AB}-\bar{BE}\right)}^{2}\to \bar{BE}=\frac{\bar{AB}}{2}+\frac{{\bar{BQ}}^{2}-{\bar{AQ}}^{2}}{2\bar{AB}}.$$

The coordinates of point *E*, $x_{E}$ and $y_{E}$, are found according to the proportional relationship.
(5)$$\{\begin{array}{c} x_{E}=x_{B}+\frac{\left(x_{A}-x_{B}\right)\bar{BE}}{\bar{AB}}\\ y_{E}=y_{B}+\frac{\left(x_{B}-y_{B}\right)\bar{BE}}{\bar{AB}} \end{array}$$

The calculation method for finding the midpoint *E* of the intersection of circle A and circle B is then used to find points *D* and *F* (as shown in Figure 17). The coordinates of points *D*, *E*, and *F* are then averaged to obtain the coordinates of point *G*, which are the coordinates of the medical device.
(6)$$\{\begin{array}{c} x_{G}=(x_{D}+x_{E}+x_{F})/3\\ y_{G}=(y_{D}+y_{E}+y_{F})/3 \end{array}$$

The above indoor positioning function [20,21,22,23] of the presented system is mainly to locate medical resources such as medical equipment and devices, medical personnel, and patients or care recipients to solve the difficulties in renting, searching, and managing medical equipment. At the same time, the positioning of medical personnel provides the function of finding personnel in case of emergencies, as well as the ability to send warning notifications when patients or care recipients enter dangerous areas and to notify caregivers to intervene for protection and treatment. With limited human resources, it can reduce the work pressure on medical staff and medical management staff and improve the efficiency and quality of care.

However, if further patient or care recipient safety management is required, for example, when a patient or care recipient enters a dangerous area of the hospital, such as a stairway area, or when a care recipient is found to be leaving the hospital through the entrance area with the driveway outside, the use of more accurate location methods must be considered and addressed. If the positioning information in these areas is not accurate enough, there is a possibility that the patient or care recipient may be injured due to inaccurate positioning information without warning notification. Therefore, this study proposes to use fingerprint positioning [41,42,43,44,45] in these dangerous areas to improve the accuracy of positioning to avoid danger to patients or care recipients.

The main process of fingerprinting positioning can be divided into four steps. (1) The signal strength data of the localization sensors in the area of the environment are collected and uploaded through the Bluetooth Hubs installed in the environment. (2) In order to train the collected signal strength data, the data must be pre-processed into the data format required by the machine learning model. Then, the data collected by the Bluetooth Hubs are combined through the time stamp of the sensor, and the signal strength data originally uploaded in a single entry are now combined into the signal strength data set received by the Hubs at the same time, providing the necessary data for the model. (3) Two machine learning models: K-nearest Neighbor (KNN) algorithm [25,46,47] and Support Vector Machine (SVM) [48], are used and compared. (4) Signal entries (data) are divided into the training data and the test data, and the RSSI signal strength is used as the feature value and the cell number as the label.

#### 3.2.3. Work Coordination Information for Medical Staff

In addition to the transmission of patient physiological information and positioning information, the presented system will also provide work coordination information among healthcare workers to facilitate real-time communication and dissemination of important information among healthcare workers. Therefore, the Publish/Subscribe of the MQTT protocol is used to divide the information exchange/transmission of medical personnel into two modes: one-to-one message transmission and group message dissemination, and to establish two different names of Group Topic and User Topic according to the different modes. The Topic is differentiated according to the different modes. In the group message dissemination mode, according to the different medical departments in the medical institution, each department is divided into different groups, and each department is given a Group Topic so that medical staff can register (Subscribe) to the Group Topic of their respective medical departments, and then they will be able to publish messages for the specific Group Topic and send the information to the designated medical departments to achieve the purpose of group message publishing. The transmission process is shown in Figure 19.

On the other hand, one-to-one messaging is mainly used to send messages from a user to a specified recipient. Although it gives a unique User Topic to the user, the same as Group Topic, and users are allowed to register for their own User Topics, each User Topic has only one user registration. Therefore, when a user needs to send a message to a specific object, he/she only needs to publish the message to the User Topic of that specific person. Since the User Topic has only one registrant, that is, the specific person, only the specific person can receive the message, thus achieving the purpose of one-to-one message transmission.

### 3.3. Message Transmission Security Protection Mechanism

When designing the protocol for IoT, the main consideration is availability and real-time, and security is not taken into account. The Transport layer of TCP or UDP does not have a security protection mechanism. Therefore, the MQTT protocol, which is very inadequate for the security protection of data [37], does not have a mechanism to protect the message content from being stolen by intentional people in the transmission process. However, in the message transmission process of this platform, the message content often contains patient information, medical data, and other information about personal privacy, and the security issue is of importance for MQTT in the presented system.

Therefore, a data security protection mechanism is designed, and the transmission flow is shown in Figure 20. When the sender sends a message, the data is reordered through the mechanism of shuffling the transmission order, and then the encryption key is used to encrypt the content of the transmitted message by XOR calculation. The encrypted message is sent to the message relay (Broker). After receiving the encrypted message, the receiving end only needs to decrypt the encrypted data content with the same key. Through this data transmission security protection mechanism, even if the message content is stolen during the transmission process, the stolen data will only be in the messy encrypted format, and the stealer cannot decrypt it to obtain the exact message content without the key.

## 4. Results

The functions of the presented medical information management system are divided into the following three parts addressed in Section 4.1, Section 4.2 and Section 4.3. Section 4.4 illustrates the message transmission security protection mechanism.

### 4.1. Management of Patient Physiological Information Monitoring

Through this management platform, healthcare workers can receive various physiological data of patients, monitor their physiological statuses at any time, and immediately handle abnormal physiological data, such as high blood pressure and fever, as shown in Figure 21. Figure 22 shows the physiological information page of the nursing station, from which the nursing staff can understand the current physiological information of the patients under their care in real time. Figure 23 shows the basic information of all patients in each ward. This page allows nursing staff to understand the current physical status of all patients in each ward and the related nursing information.

Figure 24 shows more detailed information about the patient’s personal data, the information of the wearable device (e.g., the physical address of the temperature measurement smart T-shirt Tag or the smart electronic diaper Tag), the real-time physiological data of the patient, and the curves of the historical physiological data. Figure 25a shows the display screen of real-time physiological information (body temperature) through the smartphone App with MQTT protocol, and Figure 25b shows the page for setting the alert thresholds of the body temperature and distance.

### 4.2. Management of Medical Personnel and Equipment Location Information

The location of medical devices and equipment in the hospital area can be grasped in real time, as shown in Figure 26, through the deployment of positioning devices and Bluetooth Hubs and using the circle–circle intersection technique to assist healthcare staff in management.

For location positioning in security-sensitive areas, fingerprint positioning is used. In order to verify its accuracy, the following experiments were conducted.

A 10 × 8 m^2^ indoor space was used as the experimental environment; it was divided into 49 cells, of which each is approximately 1 m^2^, as shown in Figure 27; each cell was numbered 1, 2, 3, 4, …, 49 from top to bottom and left to right. Four Bluetooth Hubs were built at the four corners of the environment (cells #1, #7, #43, and #49) in order to collect the signal strength of the sensor being positioned and the signal strength of the sensor collected by each Bluetooth Hub was used as the feature of the sensor in the cell. In this experimental environment, there are tables, chairs, computer equipment, iron cabinets, etc. The rectangular diagram in Figure 27 represents the iron cabinets, which are more likely to block the Bluetooth signal. This experimental environment is used to simulate the medical site, such as the outpatient hall and appraisal counter in the lobby (cells #3~5, #10~12, #17~19, and #24~26), the aisle area outside the ward (cells #6, #13, #20, #27, #34, #41, #48, #2, #9, #16, #23, and #30~33), the area inside the ward mostly contains beds and metal cabinets (cells #37~39 and #44~46).

In order to collect the signal strength of the sensors in each cell, a sensor was placed in the center of each cell and simulated to be worn by the user (this sensor was placed 110 cm above the ground). Each Hub collected 100 points of signal strength. A total of (49 cells ×100 points × 4 Hubs) and 19,600 samples of data were collected. The data collected by the four Bluetooth Hubs were combined through the time stamp of the sensor, and the signal strength data originally uploaded in a single entry were now combined into the signal strength data set received by the four Hubs at the same time, as shown in Figure 28, with a total of 4900 entries, providing the necessary data for the machine learning models. A total of 4900 signal entries were divided in a ratio of 9 to 1. 90% of the data were used as the training data and 10% as the test data, and the RSSI signal strength was used as the feature value and the cell number as the label. The data were trained using KNN and SVM. Figure 29 shows the accuracies and execution times of these two models. The accuracy and execution time are 0.84 and 0.171836 s for KNN, and 0.83 and 3.297741 s for SVM. 

In order to compare the accuracy of the intersection method and the accuracy of the fingerprinting positioning method, the test data were obtained by the sensor carried by a user walking from cell #2 to cell #49 (12 cells in total) illustrating by red arrows, as shown in Figure 30. One hundred points (signal samples) in each cell were collected by each of the four Hubs. A total of 4800 points (12 cells × 100 points × 4 Hubs) were collected. These data were then pre-processed and combined into 1200 entries, and each entry represents the locational signal strength data set received by the four Hubs at the same time. The coordinates of the sensor location were first calculated by performing the intersection method based on the signal strength collected from the Bluetooth Hubs. According to the actual experimental results, the average positioning accuracy of the circle–circle intersection positioning technique is 2.5 m. This is sufficient for the purpose of locating medical equipment and devices in the institution, although it is not good enough for navigation purposes. The calculated coordinates were further converted into cell numbers. The obtained cell numbers were then compared with the true location cell numbers, and the accuracy of the intersection method was only about 39%. The main reason is that each cell is about 1 square meter in size, and the intersection positioning accuracy resolution would fall out of that range sometimes. Therefore, it is found that most of the positions calculated by the intersection method fall in the surrounding cells of the correct cell, and this error is not allowed for the safety management of patients or care recipients.

On the other hand, the accuracy of KNN prediction can reach about 70~80% by using the same test data and using the above trained KNN model (using 4900 data as the training data and selecting a different number of neighbors from 1 to 10). The number of different neighbors used in the model is found to have little effect on the accuracy rate.

### 4.3. Management of Medical Staff Work Coordination Information Exchange and Transmission

Through the presented platform, healthcare workers can exchange opinions with each other and disseminate information in specific medical departments. Take bed cleaning as an example; after the physician approves the patient’s discharge, the medical staff will notify the patient that he/she can be discharged, and the used bed needs to be cleaned before it can be provided to the next patient for use. The process of bed cleaning is shown in Figure 31, in which the nursing staff needs to notify the cleaning staff to clean the bed first, then the nursing staff needs to be notified again to make the bed and set up the medical equipment after the cleaning staff has finished, and finally, the nursing staff notifies the bed manager to complete the bed cleaning procedure. Through this platform, the bed cleaning staff can create a Topic for bed cleaning, under which they can notify or be notified of the bed cleaning and understand the usage status of the bed (as shown in Figure 32).

### 4.4. Message Transmission Security Protection Mechanism

In this study, we encrypted the contents of messages when they were transmitted by users through the message security protection mechanism. When we intercepted the message packets sent by users to MQTT Topics through the packet capture software (WireShark), we found that when the message security protection mechanism was not used, the contents of the transmitted messages were obviously presented directly in the packet contents (Figure 33), and sensitive personal information could be easily stolen. When the message security protection mechanism is used to encrypt the message contents, the intercepted packets do not show the original message contents, but the encrypted irregular garbage code (Figure 34), and the decryption key is needed to restore the message to the correct contents.

## 5. Discussion

The results of (a) the management of patient physiological information monitoring, (b) management of medical personnel and equipment location information, (c) management of medical staff work coordination information exchange and transmission, and (d) message transmission security protection mechanism show the presented system works properly and is able to provide necessary service functions to medical/healthcare institutions to reduce the loading of medical staff. Since Bluetooth is used as a medium for data transmission in the system, integrating additional medical or wearable devices into the presented system in the future is feasible as long as they are Bluetooth-based devices and their data protocols are accessible. Such devices could be ear thermometers, sphygmomanometers, oximeters, intravenous drip controllers, electrocardiography (ECG) units, electronic stethoscopes, etc. Moreover, the Bluetooth Hub at the entrance of the ward in Figure 7 collects the data from the smart T-shirts and smart diapers through Bluetooth broadcasting. There can be many Bluetooth-based devices in the ward. The use of Bluetooth-based smart T-shirts and smart diapers in the institutions provides, for the first time, the ability for long-term, continuous monitoring of patients or care recipients. Since the variation of temperature or humidity is slow, these data are broadcast every second. The size of the data packet is about 32–50 Bytes. The tags of smart T-shirts and smart diapers not only can collect physiological data from wearable devices (T-shirts and diapers) but also can be used for indoor positioning as sensors. Therefore, the person who carries or wears it can be tracked or located. If security issues are concerned, when the person enters a dangerous area, a warning message can be sent to his/her caregiver to intervene for protection or treatment.

For the purpose of finding lost or misplaced medical equipment and devices in the institution, the indoor positioning function using the intersection technique provides sufficient location information to locate medical equipment and devices, although it may not be good enough for navigation purposes. However, for location positioning in security-sensitive areas, fingerprint positioning was adopted in this study to provide better location positioning resolution to improve the accuracy of positioning and to avoid danger to patients or care recipients. Unlike the experimental scenario proposed by Ruan et al. [49], where 36 Hubs were deployed in a space of 300 m^2^ with a grid size of 0.6 m × 0.6 m and 868 reference points, the experimental site in this study is 49 m^2^ with a grid size of 1 m × 1 m and 49 reference points and has only four Hubs with a spacing of 7 m. Two machine learning models, KNN and SVM, were tested for fingerprint positioning. It was found that the accuracy of KNN was better than that of SVM. However, the difference between them was not significant, but the time of training SVM was more than 18 times that of KNN. While looking at the positioning error of [49] using the weighted KNN, we found that the average error is 4.04 m. The KNN used in this study obtained an accuracy of 0.84, i.e., 84% of errors were within 1 m. When compared with the intersection method, the KNN fingerprinting positioning method can effectively and significantly improve the accuracy of localization, which is very helpful for the safety management of patients or care recipients. However, to perform fingerprinting positioning, it is necessary to collect enough locational features in the area to be localized beforehand to achieve the above accuracy, which is a considerable burden and causes a lot of localization setup costs. Therefore, this study suggests that the fingerprinting positioning method should be carried out only for the dangerous areas, while most of the areas in the hospital should still use the circle–circle intersection method for indoor positioning so that the cost of localization can be reduced and the time of localization can be shortened, which is more in line with the actual needs of localization.

The use of a security protection mechanism for MQTT ensures the privacy of personal sensitive medical information from being stolen. Therefore, the patient’s physiological information and the medical staff’s work coordination information can be protected. Although there exists popular social software, such as LINE, QQ, Skype, Facebook messenger, etc., can provide the function of the management of medical staff work coordination information exchange, the privacy issue in the medical information still needs to be taken care of. The developed functions with the proposed secured MQTT mechanism in this paper fulfill the needs for the healthcare information and medical resource management system.

In order to compare the differences between the presented system and similar systems such as the ones presented in [24,50], several key functionalities are explored: sensors, communication, indoor positioning, MQTT, and medical staff work coordination information exchange. Table 3 lists the differences. The system of [50] developed a Bluetooth-based ECG sensor, while the system presented in this paper used smart diapers and smart T-shirts to measure temperature and diaper humidity. The system of [24] did not discuss any implementation of sensors. All three systems used Bluetooth to connect sensors and the data collector (coordinator node, PDA, or Bluetooth Hub/smartphone). Only the system presented in this paper provides functionalities of indoor positioning, medical staff work coordination information exchange, and secure MQTT.

## 6. Conclusions

In today’s healthcare manpower shortage environment, the complexity of patient care and equipment management has put each healthcare worker under considerable work pressure, and there are frequent reports of overworked healthcare workers, highlighting the increasing demand for human resources in healthcare facilities. Therefore, in this study, the common work contents of healthcare workers have been integrated into a medical information management system through the Internet of Things technology to allow healthcare workers to receive work information in real time. An encryption mechanism for information protection has also been proposed to avoid the leakage of medical-related information. The experimental results show that healthcare workers can monitor the physiological condition of patients, locate and track medical equipment, exchange information and communicate with other healthcare workers at any time, and accelerate the acquisition of coordinated information for healthcare workers effectively, improving work efficiency and reducing the work pressure of healthcare workers.

## Figures and Tables

**Figure 1 sensors-23-05389-f001:**
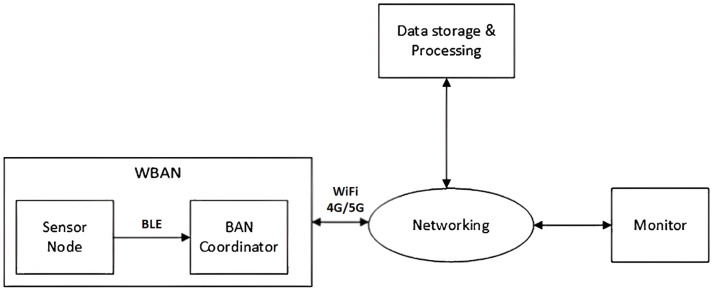
Remote Health Monitoring Systems based on Wireless Body Area Network.

**Figure 2 sensors-23-05389-f002:**
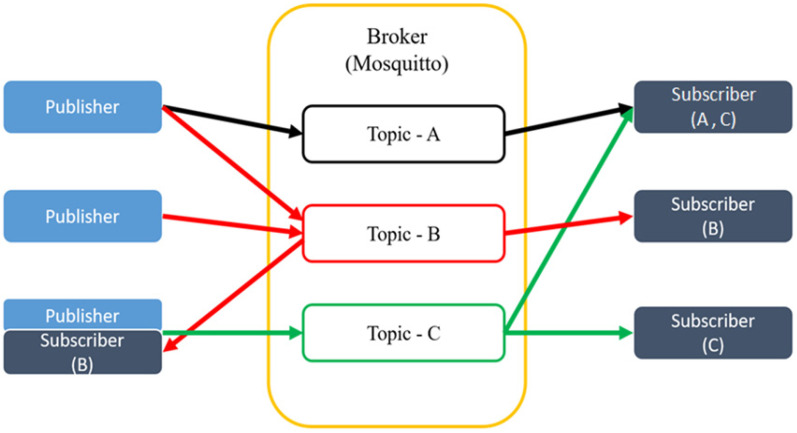
The data flow of transmission for Publishers and Subscribers.

**Figure 3 sensors-23-05389-f003:**
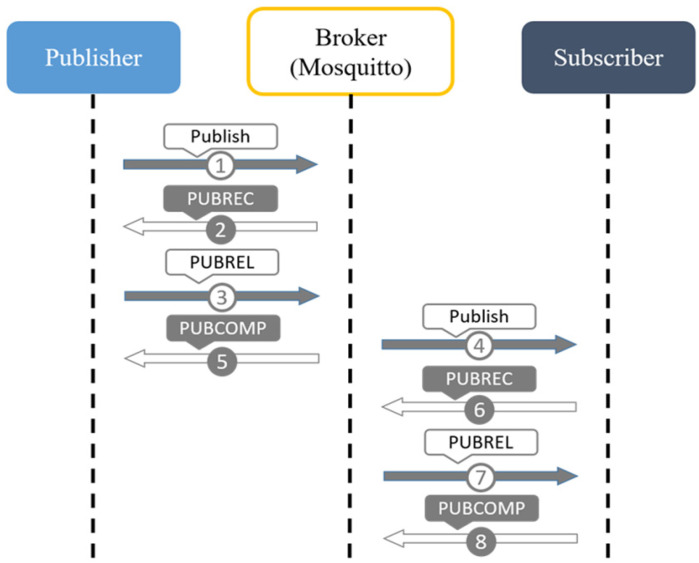
The transmission protocol of QoS Level 2.

**Figure 4 sensors-23-05389-f004:**
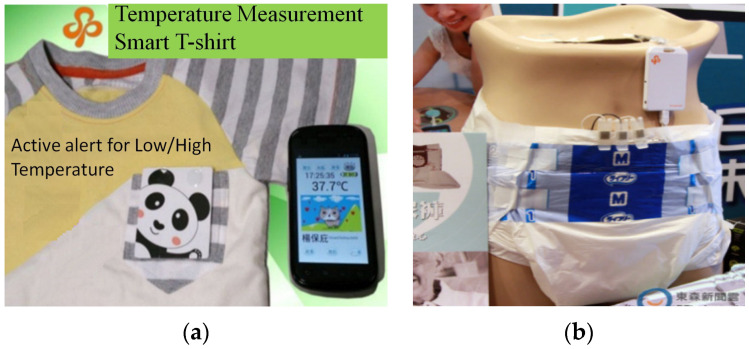
(**a**) Temperature measurement smart T-shirt and (**b**) smart adult diaper.

**Figure 5 sensors-23-05389-f005:**
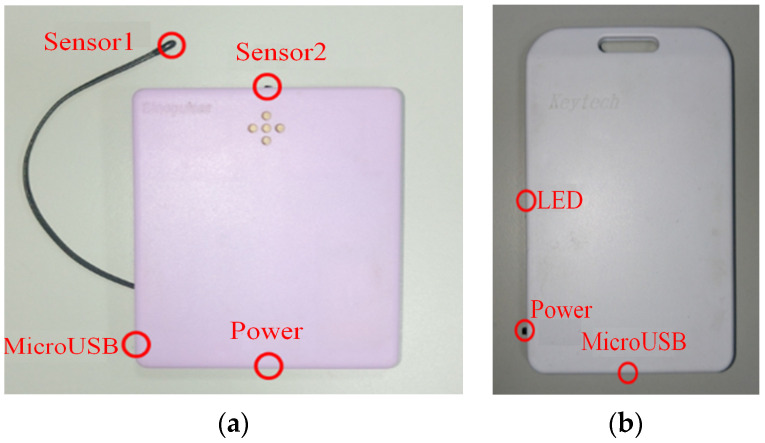
(**a**) Tag of temperature measurement smart T-shirt and (**b**) tag of smart adult diaper.

**Figure 6 sensors-23-05389-f006:**
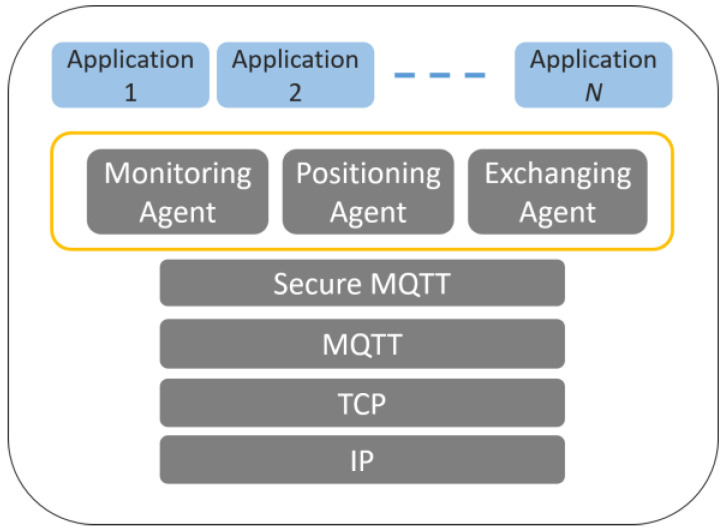
Protocol hierarchy map of the platform.

**Figure 7 sensors-23-05389-f007:**
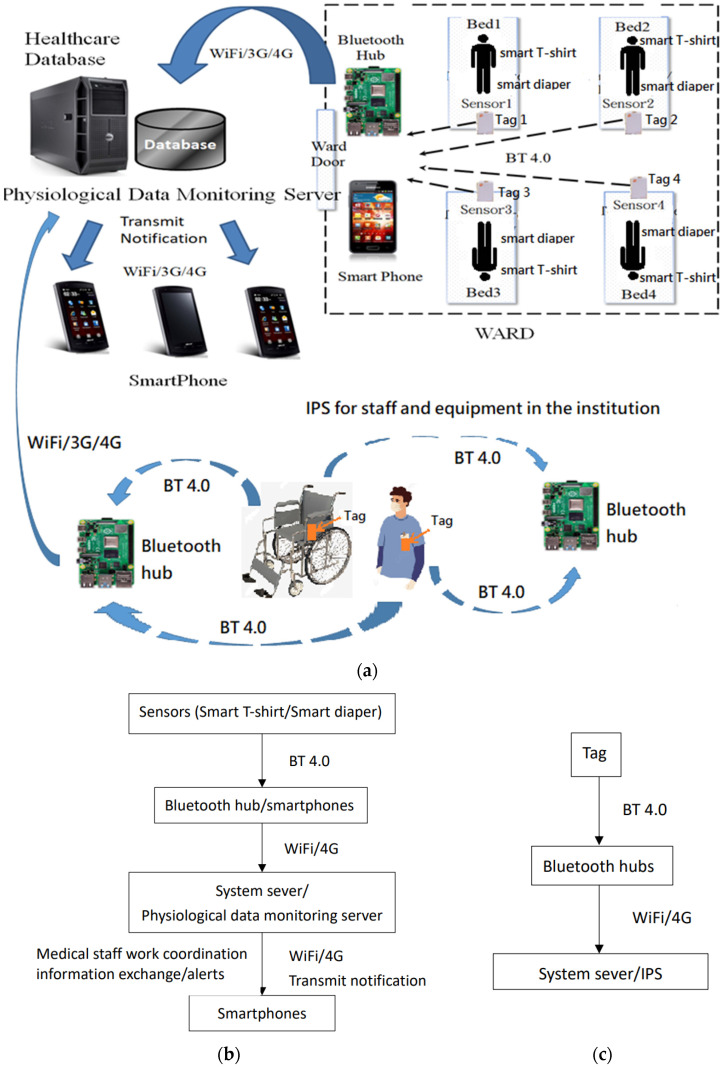
(**a**) System usage scenarios, (**b**) workflow chart for smart sensors, (**c**) workflow chart for IPS.

**Figure 8 sensors-23-05389-f008:**
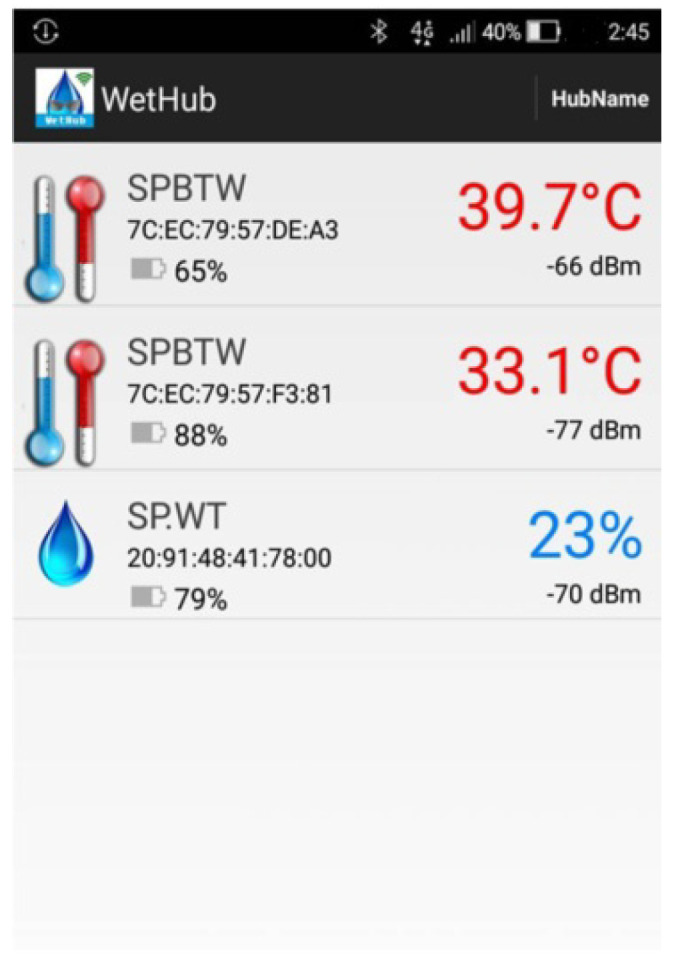
Humidity information collection screen.

**Figure 9 sensors-23-05389-f009:**
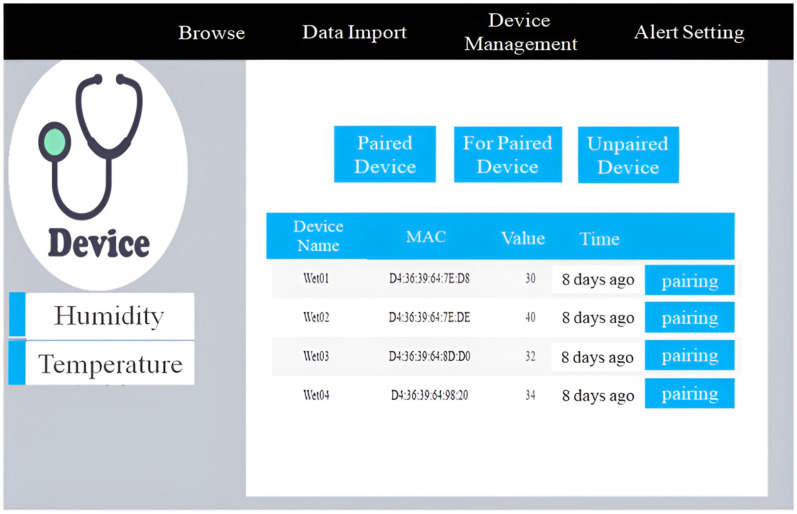
Patient sensing equipment pairing.

**Figure 10 sensors-23-05389-f010:**
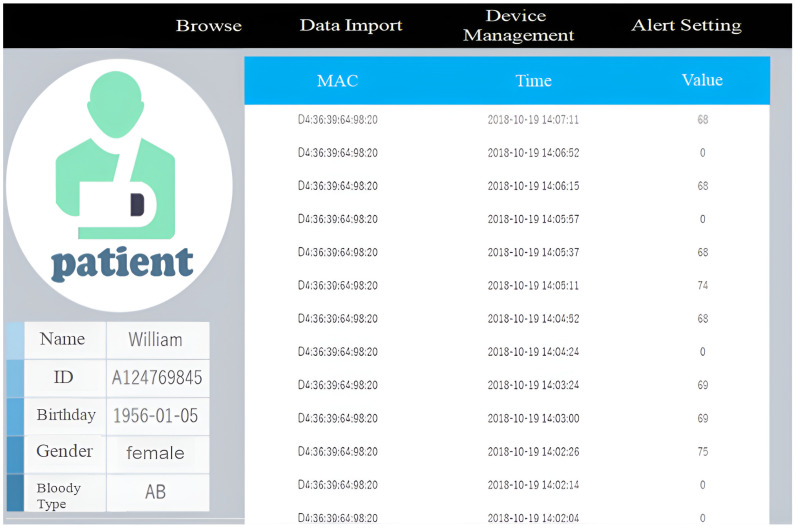
The collected humidity information displayed on back-end physiological information collection server.

**Figure 11 sensors-23-05389-f011:**
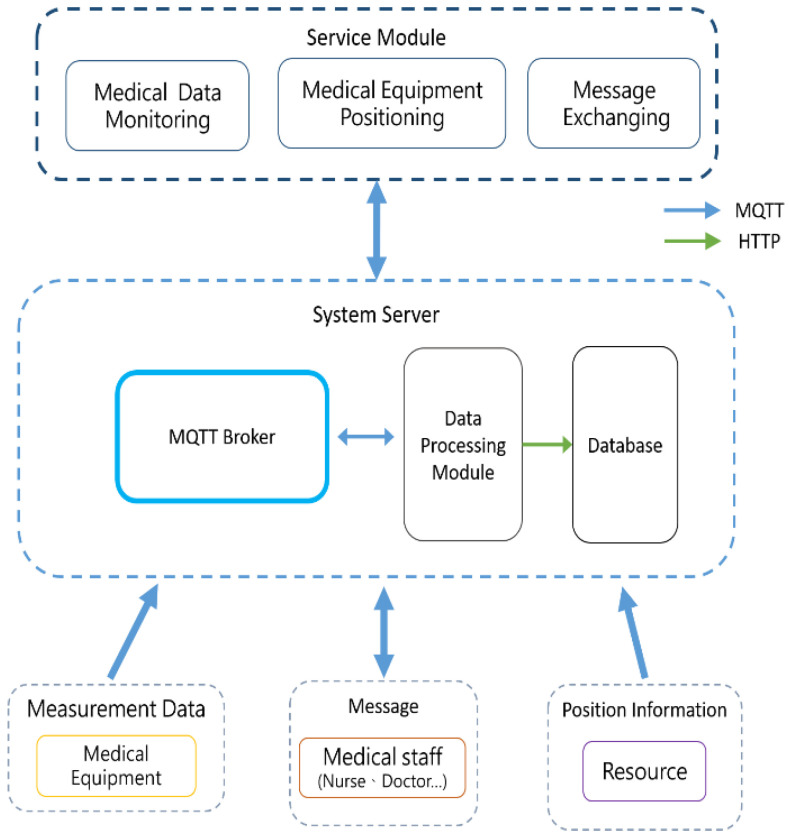
System architecture.

**Figure 12 sensors-23-05389-f012:**
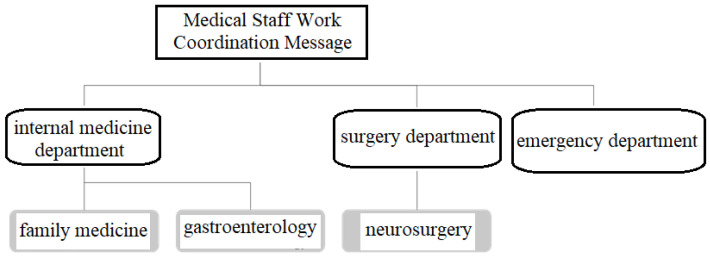
Schematic diagram of hierarchical structure of naming.

**Figure 13 sensors-23-05389-f013:**
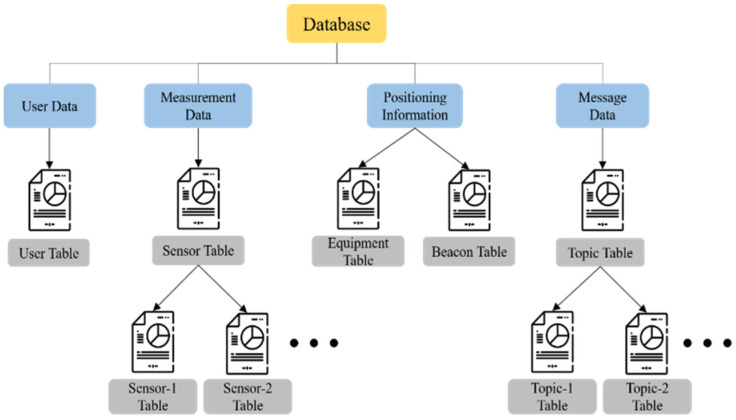
Schematic diagram of data table.

**Figure 14 sensors-23-05389-f014:**
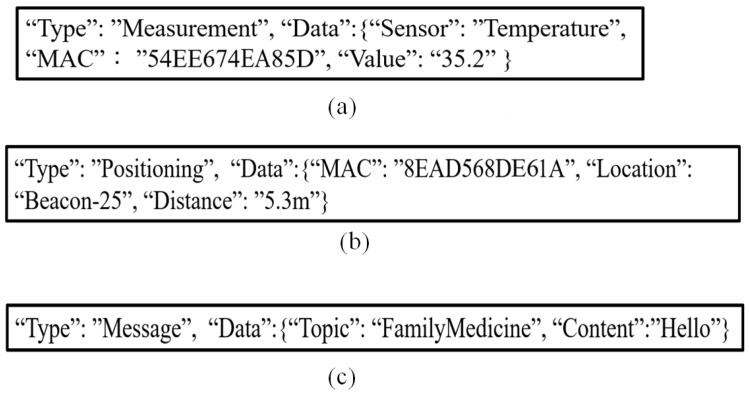
(**a**) Sensor data format, (**b**) positioning data format of medical equipment, and (**c**) data format of message exchange.

**Figure 15 sensors-23-05389-f015:**
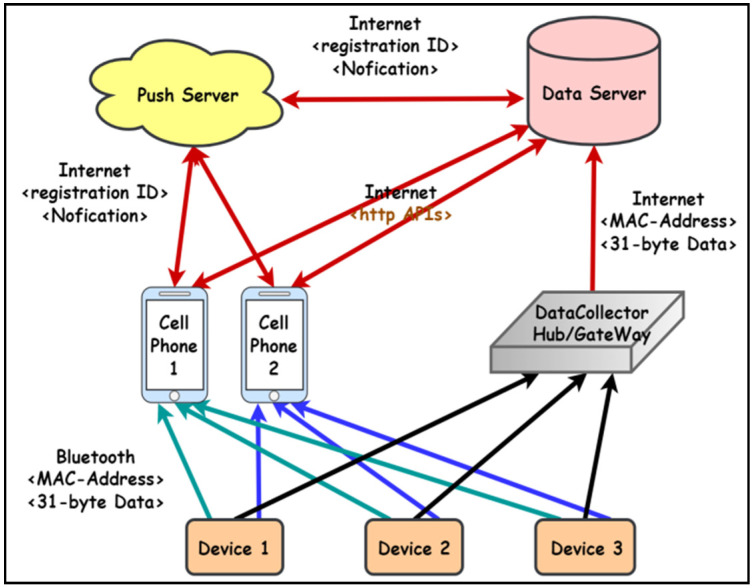
Push message data transmission process.

**Figure 16 sensors-23-05389-f016:**
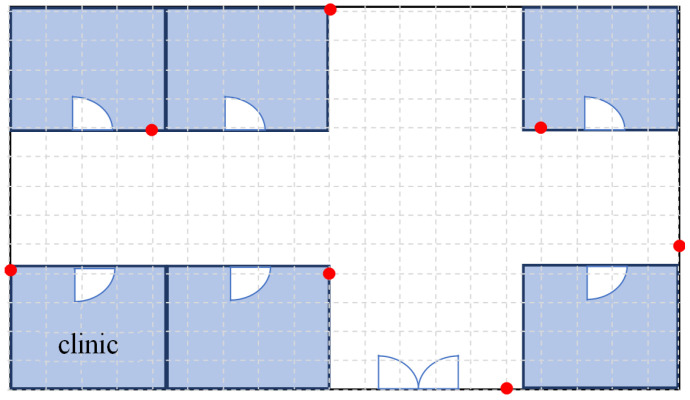
Conceptual diagram for deployment of Bluetooth Hub.

**Figure 17 sensors-23-05389-f017:**
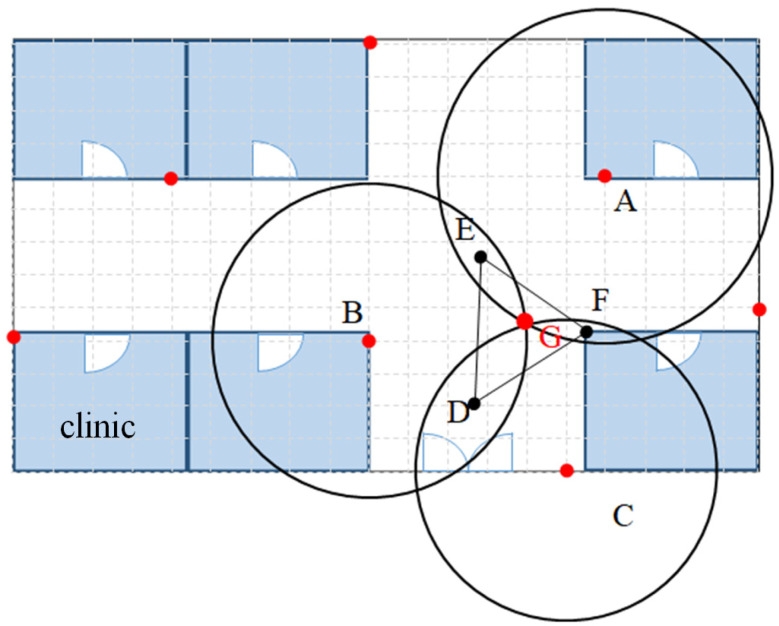
Intersection points of three circles for centroid.

**Figure 18 sensors-23-05389-f018:**
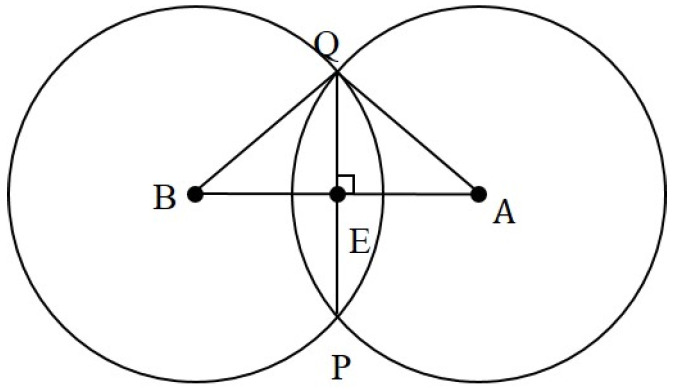
Intersection points of two circles.

**Figure 19 sensors-23-05389-f019:**
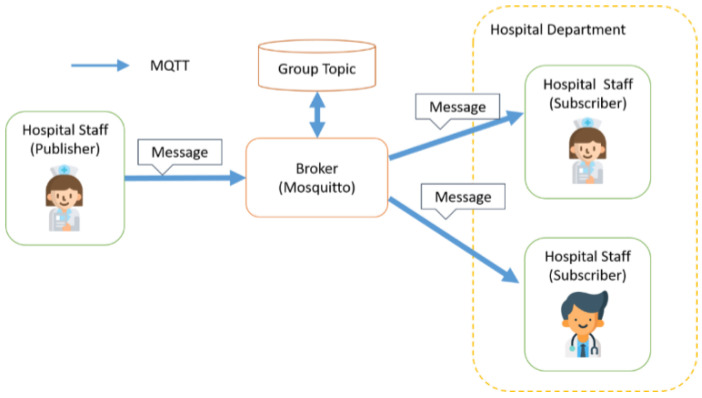
Flowchart of group message transmission.

**Figure 20 sensors-23-05389-f020:**
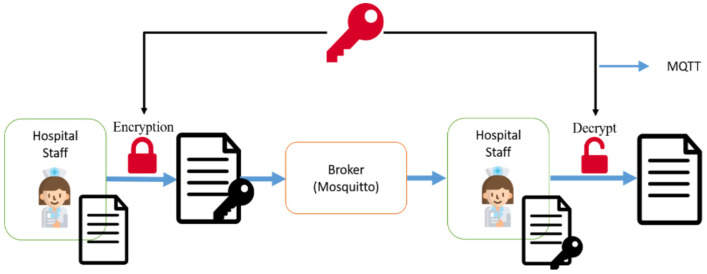
Flowchart of message encryption and decryption.

**Figure 21 sensors-23-05389-f021:**
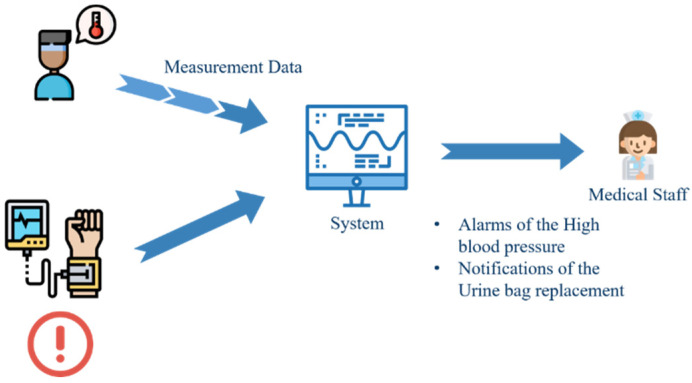
Management of patient physiological information monitoring.

**Figure 22 sensors-23-05389-f022:**
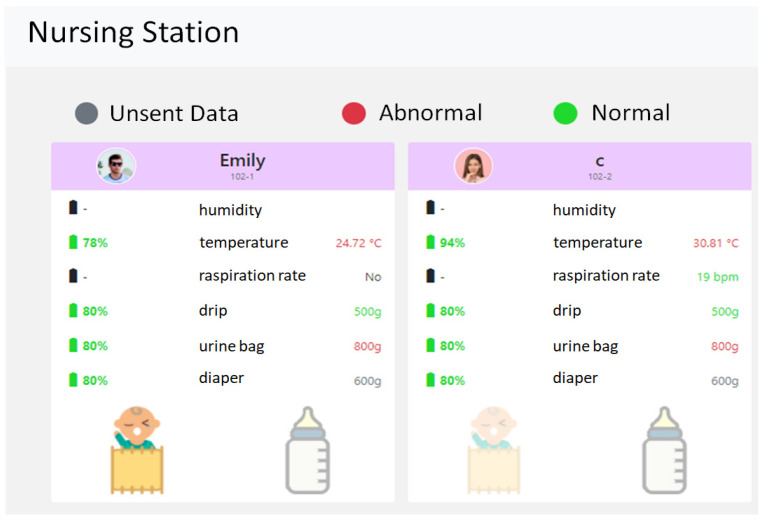
Monitoring page of patient physiological information at nursing station.

**Figure 23 sensors-23-05389-f023:**
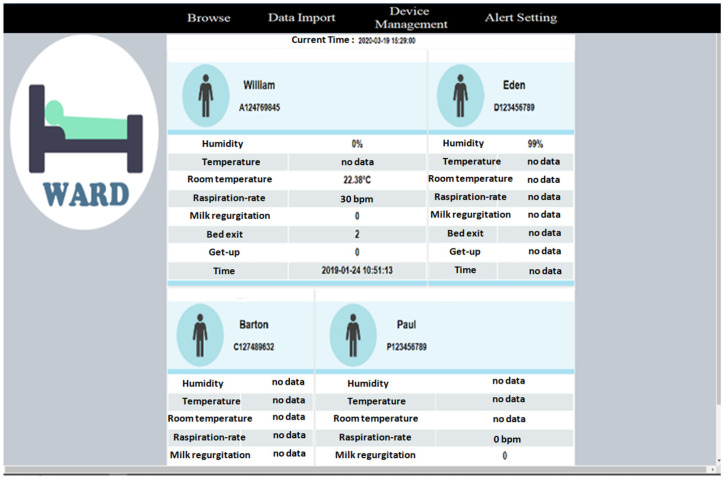
Basic and physiological information for all patients in the ward.

**Figure 24 sensors-23-05389-f024:**
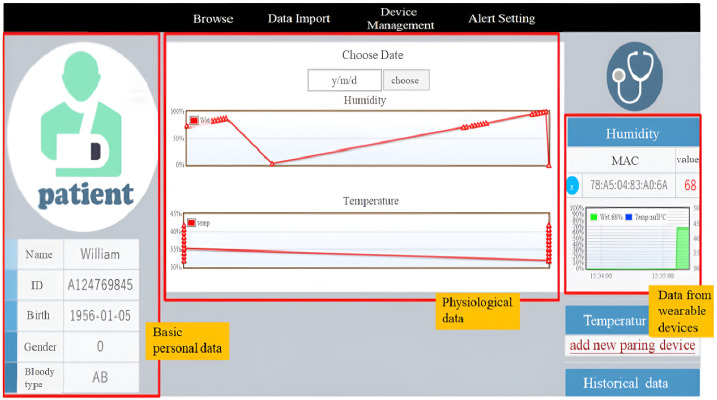
Patient’s basic information, data from wearable devices, physiological information.

**Figure 25 sensors-23-05389-f025:**
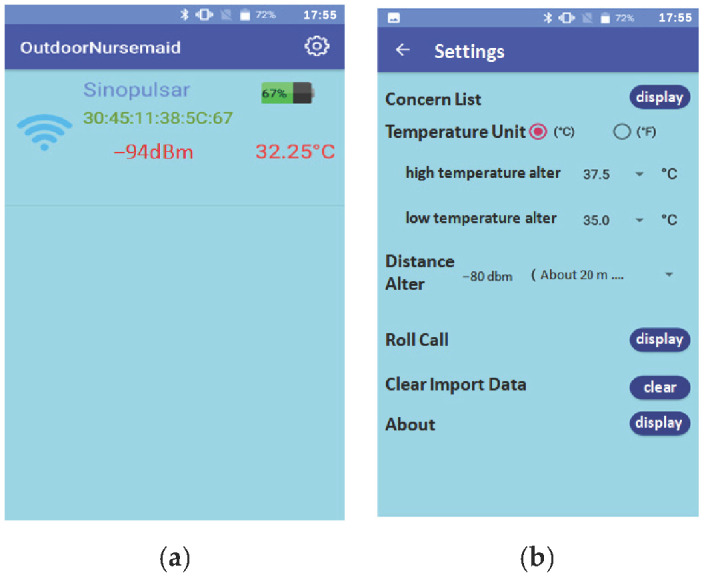
(**a**) Real-time temperature and (**b**) temperature/distance alert setting.

**Figure 26 sensors-23-05389-f026:**
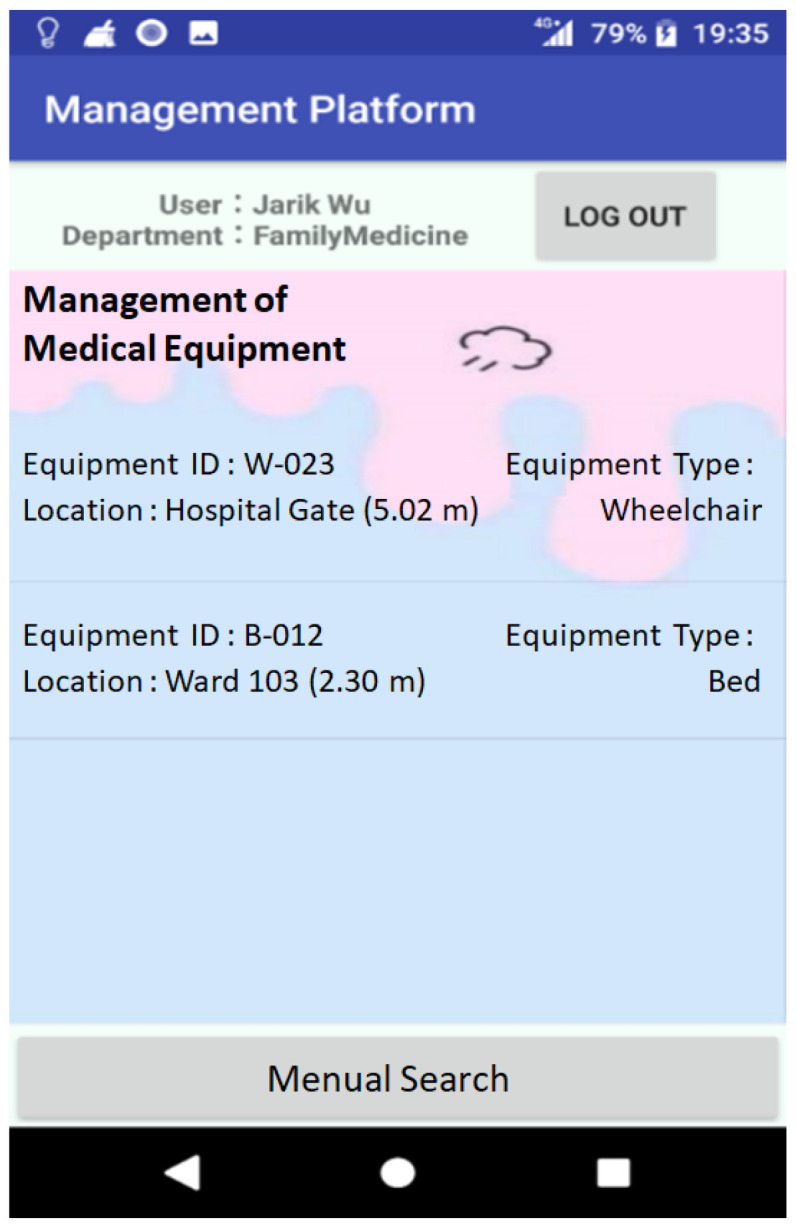
Medical Equipment Positioning information.

**Figure 27 sensors-23-05389-f027:**
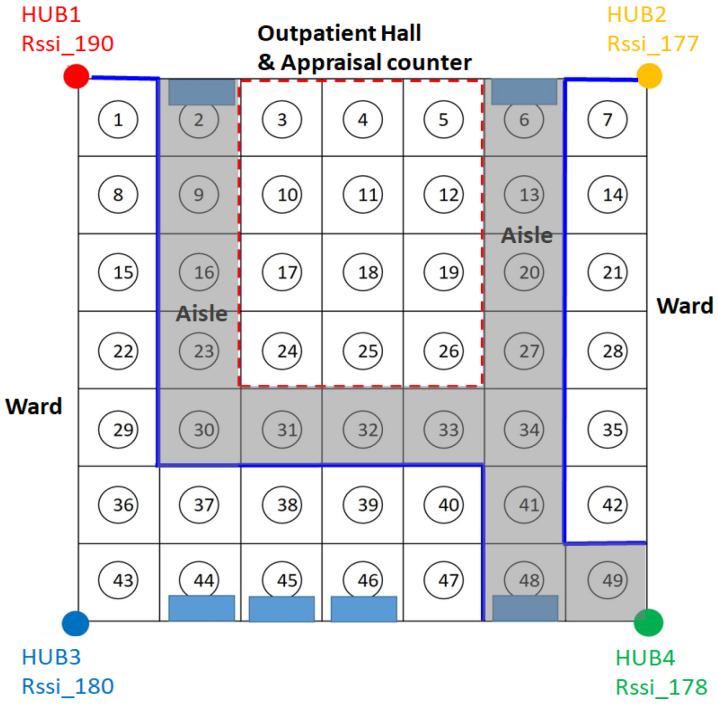
Experimental environment for fingerprint positioning.

**Figure 28 sensors-23-05389-f028:**
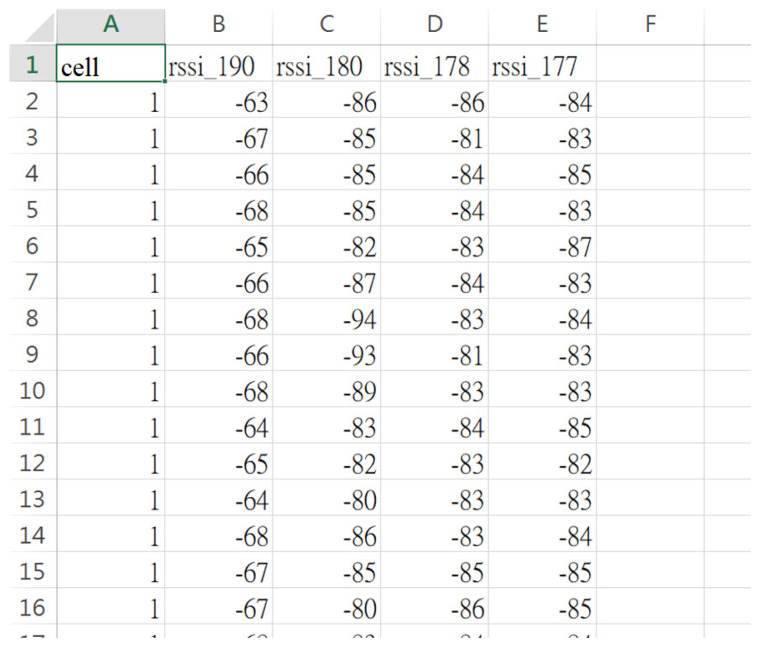
Training data of cell 1 after pre-processing.

**Figure 29 sensors-23-05389-f029:**
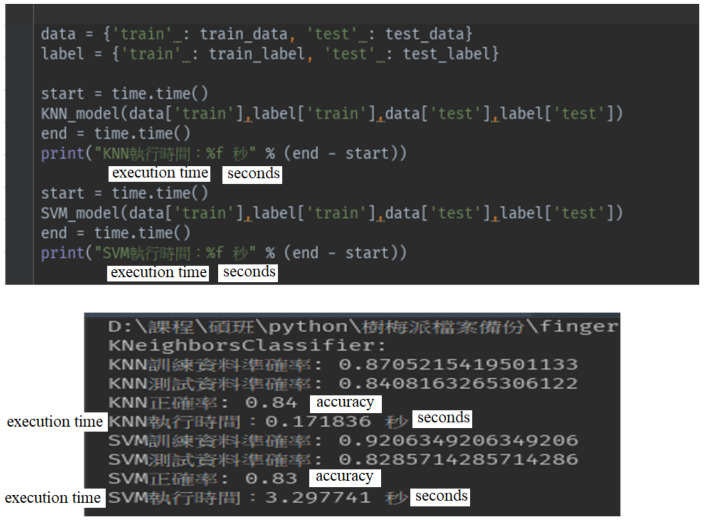
Results of KNN and SVM models.

**Figure 30 sensors-23-05389-f030:**
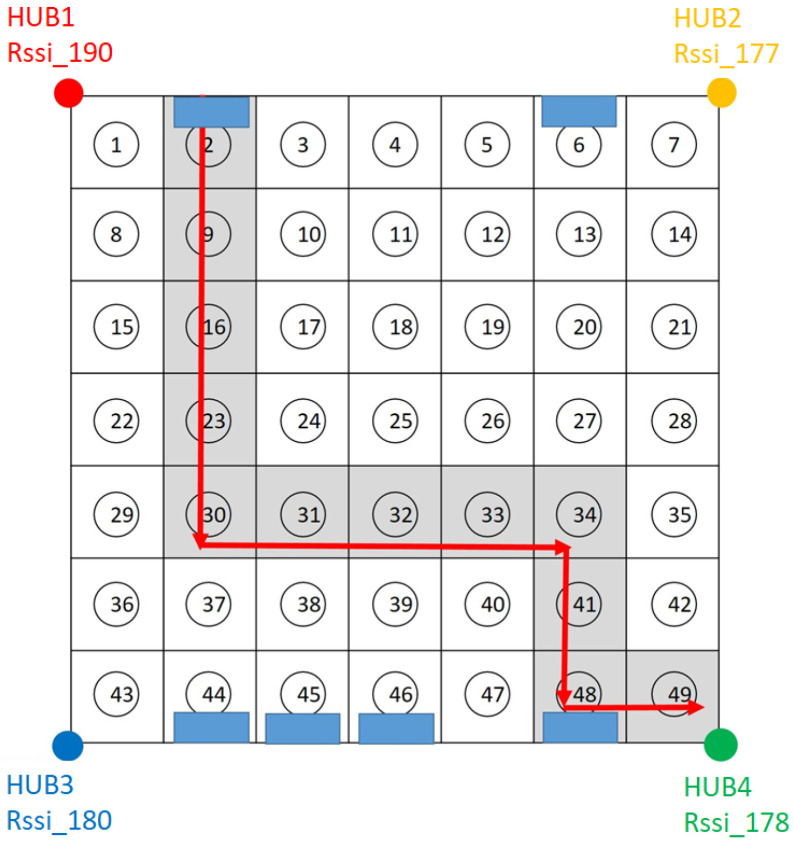
Walking path.

**Figure 31 sensors-23-05389-f031:**
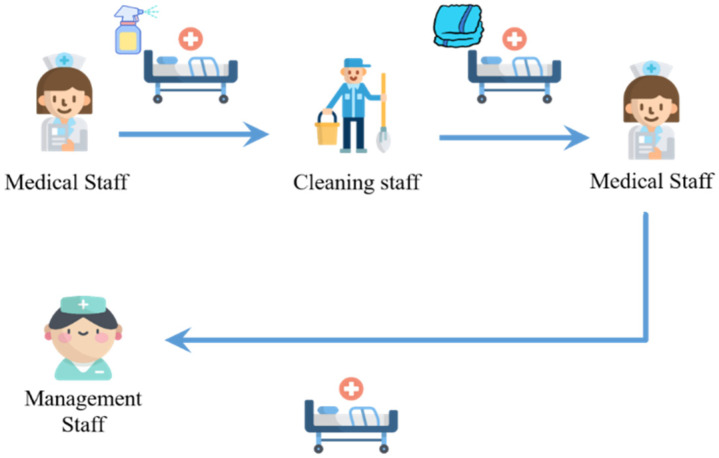
Bed cleaning flow chart.

**Figure 32 sensors-23-05389-f032:**
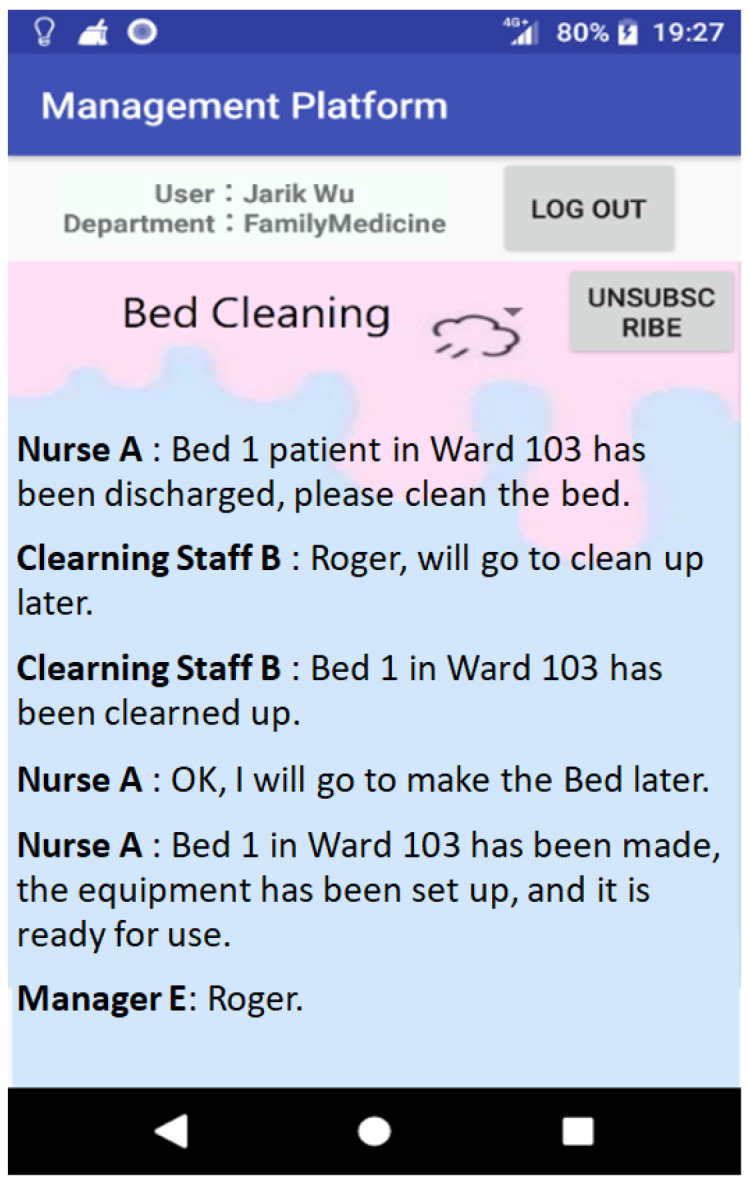
Medical staff’s work coordination information.

**Figure 33 sensors-23-05389-f033:**
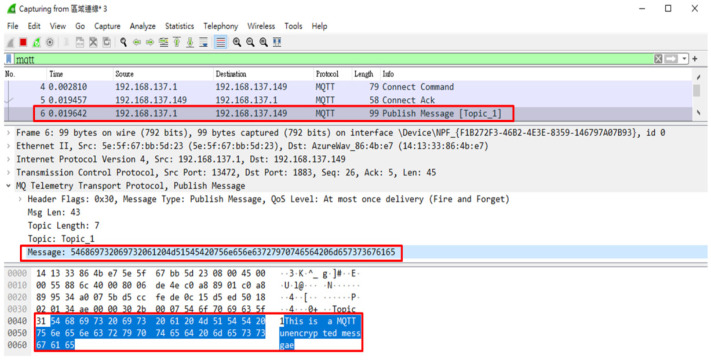
Contents of unencrypted message packets.

**Figure 34 sensors-23-05389-f034:**
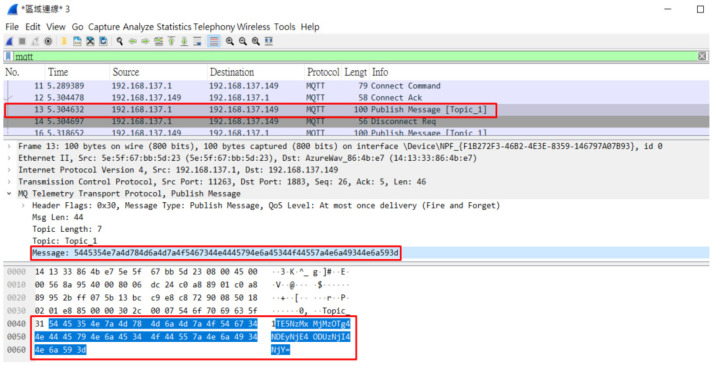
Contents of encrypted message packets.

**Table 1 sensors-23-05389-t001:** Topic Name Categories.

Medical Information Categories	Topic Names
Patient physiological monitoring information	Measurement/…
Medical staff and equipment positioning information	Positioning/…
Medical staff work coordination information	Message/…

**Table 2 sensors-23-05389-t002:** Theme names of each department.

Department	Topic Name
Internal medicine	Message/Medicine
Surgery	Message/Surgery
Family medicine	Message/Medicine/Family Medicine
Neurosurgery	Message/Surgery/Neurosurgery

**Table 3 sensors-23-05389-t003:** Functionalities of three different systems.

Functionalities	Ref. [50] (the Year 2009)	Ref. [24] (the Year 2020)	This Paper
Sensors	Bluetooth-based ECG	Literature review, no implementation of any sensors	Smart diaper Smart T-shirt
Communication	Sensor data to a coordinator node via Bluetooth USB used for connection between PC (mobile phone) and coordination node GPRS for mobile phones, Internet for PC	Sensor data to a PDA via BLE WiFi/4G used for connection between PDA and cloud or medical server	Sensor data to a Bluetooth Hub or smartphone via BLE, WiFi/4G used for connection between Bluetooth Hub/smartphone and PC (server)
Indoor positioning	X	X	√
MQTT	X	X	Secure MQTT
Medical staff work coordination information exchange	X	X	√

Notes: X means “the function is not available or not discussed”, √ means “the function is provided”.

## Data Availability

Not applicable.

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
