# Peer review of "Bluetooth-Based Healthcare Information and Medical Resource Management System"

_sensors, 2023, doi:10.3390/s23125389_

Round 1
Reviewer 1 Report
The authors have presented a Bluetooth-based Healthcare Information and Medical Resource Management System. In this paper, they have used a healthcare information and medical resource management platform utilizing wearable devices, physiological sensors, and an indoor positioning system (IPS). This platform provides medical healthcare information management based on the physiological information collected by wearable devices and Bluetooth data collectors.
The paper is written well. However, I recommend discussing related works of the year 2022 in the introduction section.
Further, the contributions of the work should be listed pointwise in the introduction section.
Author Response
- The following paragraph has been added in the Introduction to discuss related works [13]-[15][17] of the year 2022.
“Adbulmalek et al. [13] provided a systematic literature review on recent studies of IoT-based healthcare-monitoring systems, compared various systems’ effectiveness, efficiency, data protection, privacy, security, and monitoring, and discussed the challenges and open issues regarding security and privacy and Quality of Service (QoS). Heaney et al. [14] presented a custom-built, end-to-end ECG capturing prototype system. This system utilizes an AD8232 microchip (Analog Devices, USA) as the analog front-end which is then fed into an Arduino (MKR1010, Arduino, Italy) which offers both Bluetooth and WiFi connectivity to a smartphone or another external device for the remote monitoring of the patients. The system is also equipped with temperature and specific oxygen (SpO2) sensors, and can display an ECG signal/SpO2/temperature on a local display screen, on a custom mobile application through Bluetooth, and on a server through WiFi. Tang et al. [15] proposed a 5G-based architecture for smart healthcare information infrastructure, which optimized the latest technical architecture standardized by 3gpp and ETSI about MEC 5G Integration under a smart healthcare scenario. A new network element iGW (industry gateway) as the core was defined, and a smart healthcare dedicated cloud platform iMEP (industry multi-access edge platform) was also introduced.”
“Azbeg et al. [17] presented a secure healthcare system that integrates IoT with Blockchain to support remote patient monitoring, especially when it comes to chronic diseases that need regular monitoring. The security was ensured by using the re-encryption proxy together with Blockchain to store hash data via IPFS (Inter Planetary File System).”
- Adbulmalek, S.; Nasir, A.; Jabbar, W.A.; Almuhaya, M.A.M.; Bairagi, A.K.; Khan, M.A.; Kee, S.H.. IoT-Based Healthcare-Monitoring System towards Improving Quality of Life: A Review. Healthcare 2022, 10, 1993.
- Heaney, J.; Buick, J.; Hadi, M.U.; Soin, Navneet. Internet of Things-Based ECG and Vitals Healthcare Monitoring System. Micromachines 2022, 13, 2153.
- Tang, X.; Zhao, L.; Chong, J.; You, Z.; Zhu. L.; Ren, H.; Shang, Y.; Han, Y.; Li, G. 5G-Based Smart Healthcare System Designing and Field Trial in Hospitals. IET Communications 2022, 16, 1-13
- Azbeg, K.; Ouchetto, O.; Andaloussi, S.J. BlockMedCare: A Healthcare System Based on IoT, Blockchain and IPFS for Data Management Security. Egyptian Informatics Journal 2022, 23, 329-343.
- The main contributions of the presented work have been listed pointwise in the Introduction section as follows.
“The main contributions of this paper are summarized as follows.
(1) This paper integrates physiological information sensing technology and wearable devices with mobile positioning technology in an Internet of Things-based medical care and medical resource management platform.
(2) This IoT platform uses wearable devices and physiological information sensing technology to perform real-time patient physiological information monitoring and medical care information management, and uses a proposed indoor positioning tracking to establish a safe activity area for the care recipient, allowing the caregiver to understand the patient's physiological information and movement, assisting and reducing the caregiver's burden and providing an additional layer of protection mechanism for the care recipient.
(3) The IoT platform combines indoor positioning tracking for hospital equipment and device management, solving the problem of medical equipment rental management.
(4) The IoT platform provides a real-time information exchange function, allowing medical staff and medical management staff to share medical information to improve the efficiency of medical institutions.
(5) To reduce the workload of nursing and medical management staff and improve the efficiency and quality of care with limited human resources.”
Reviewer 2 Report
The authors proposed a bluetooth-based healthcare management system. This reviewer feels that the work is solid and can be accepted after the following comments be addressed:
-The performance comparison with similar approaches (such as the following one) is missing. Please assign a Section for performance comparison with similar approaches in a Table-based format.
-Y. Zhang et al., "Bluetooth-Based Sensor Networks for Remotely Monitoring the Physiological Signals of a Patient", IEEE Transactions on Information Technology in Biomedicine, 2009.
-https://www.ncbi.nlm.nih.gov/pmc/articles/PMC7313288/
-The quality of some figures (such as Figs. 6 and 7) must be improved.
Author Response
- The comparison table is as follows, and the following paragraph is added at the end of the Discussion section.
“In order to compare the differences between the presented system and similar systems such as the ones presented in [47] and [48], several key functionalities are explored: sensors, communication, indoor positioning, MQTT, and medical staff work coordination information exchange. Table 3 lists the differences. The system proposed in [47] developed a Bluetooth-based ECG sensor, while the system presented in this paper used smart diapers and smart T-shirts to measure temperature and diaper humidity. The system in [48] did not discuss any implementation of sensors. All three systems used Bluetooth to connect sensors and the data collector (coordinator node, PDA, or Bluetooth Hub/smartphone). Only the system presented in this paper provides functionalities of indoor positioning, medical staff work coordination information exchange, and secure MQTT.”
Table 3 Functionalities of three different systems
|
functionalities |
Ref. [47] (the year 2009) |
Ref. [48] (the year 2020) |
This paper |
|
sensors |
Bluetooth-based ECG |
Literature review, no implementation of any sensors. |
l Smart diaper, l smart T-shirt |
|
communication |
Sensor data to a coordinator node via Bluetooth, USB used for connection between PC (mobile phone) and coordination node, GPRS for mobile phone, Internet for PC. |
Sensor data to a PDA via BLE, WiFi/4G used for connection between PDA and cloud or medical server. |
Sensor data to a Bluetooth hub or smartphone via BLE, WiFi/4G used for connection between Bluetooth hub/smartphone and PC (server). |
|
Indoor positioning |
X |
X |
V |
|
MQTT |
X |
X |
Secure MQTT |
|
Medical staff work coordination information exchange |
X |
X |
Ö |
Notes: X means “the function is not available or not discussed,” V means “the function is provided.”
[47] Zhang, Y. and Xiao, H. Bluetooth-Based Sensor Networks for Remotely Monitoring the Physiological Signals of a Patient. IEEE Transactions on Information Technology in Biomedicine 2009, 13, 1040-1048
[48] Fourati, L.C. and Said, S. Remote Health Monitoring Systems Based on Bluetooth Low Energy (BLE) Communication Systems. The Impact of Digital Technologies on Public Health in Developed and Developing Countries, 2020 May 31, 12157, 41-54.
- The quality of Figs. 6 and 7 has been upgraded.
Reviewer 3 Report
This paper presents a healthcare information and medical resource management platform 11 utilizing wearable devices, physiological sensors, and an indoor positioning system. There is a certain amount of work, but the following problems need to be improved.
1 The introduction lacks the challenges, motivation, and contribution of the paper.
2 Lack of architecture and work flow chart of the overall work and structure of the system.
3 The method part of the article is relatively scattered, more like a technical specification, lacking in academic.
4 The relevant formula lacks explanation.
5 The logic of the article needs further improvement.
Author Response
- Modifications have been made in the Introduction to discuss the challenges, motivations, and contributions of this paper.
Challenges/Motivations:
- As the proportion of the elderly population continues to rise, the demand for medical care escalates and the workload of healthcare workers increases. This, coupled with the declining fertility rate, has resulted in a declining workforce and has highlighted the shortage of medical manpower.
- Medical devices and equipment such as wheelchairs and crutches are available for rent to people/patients, but people often forget to return them to their original place after using them, causing great problems for medical staff in the management of medical supplies and increasing the workload of medical staff.
- In addition to patient care, medical staff in medical institutions also need to manage medical equipment and supplies. Bluetooth broadcasting signals from Bluetooth beacons placed on medical devices and equipment can also be collected to locate these medical devices through the Bluetooth hub equipment deployed in medical institutions. This will reduce the inconvenience of device management and improve management efficiency.
- In the maintenance of medical equipment, management staff need to be quickly informed of the maintenance time and status of each medical equipment. For example, in the case of bed management, the statuses of beds, such as not-in-use, in-use, and clearing beds, are required to be updated immediately, so that medical staff can clearly have those empty beds for new patients. During the period between patient check-out and bed cleaning, it is often necessary for nursing staff and bed cleaning staff to provide each other with real-time information on the status of beds to improve the overall speed of the bed cleaning process. In the past practice of using telephone contact for bed cleaning, nursing staff may forget, due to their busy work schedule, to notify bed cleaners in time when patients check out, which will lead to delays in bed cleaning progress.
- The MQTT protocol lacks a security protection mechanism. An information security protection method during the information transmission process is also proposed to tackle the shortcomings of the MQTT protocol in information security protection.
- If further patient or care recipient safety management is required, for example, when a patient or care recipient enters a dangerous area of the hospital, such as a stairway area, or when a care recipient is found to be leaving the hospital through the entrance area with the driveway outside, the positioning information in these areas provides warnings to keep the patient or care recipient from being injured.
Contributions:
(1) This paper integrates physiological information sensing technology and wearable devices with mobile positioning technology in an Internet of Things-based medical care and medical resource management platform.
(2) This IoT platform uses wearable devices and physiological information sensing technology to perform real-time patient physiological information monitoring and medical care information management and uses a proposed indoor positioning tracking to establish a safe activity area for the care recipient, allowing the caregiver to understand the patient's physiological information and movement, assisting and reducing the caregiver's burden and providing an additional layer of a protection mechanism for the care recipient.
(3) The IoT platform combines indoor positioning tracking for hospital equipment and device management, solving the problem of medical equipment rental management.
(4) The IoT platform provides a real-time information exchange function, allowing medical staff and medical management staff to share medical information to improve the efficiency of medical institutions.
(5) To reduce the workload of nursing and medical management staff and improve the efficiency and quality of care with limited human resources.”
- The architecture and workflow chart of the overall work and structure of the system are modified in Fig. 4(a)~(c). Figure 4(a) shows the system usage scenarios, where care recipients wear smart T-shirts/diapers in the ward, and the Bluetooth hub at the entrance of the ward and the APP-Hub (either BT or WiFi) on the smartphone are running the information collection program to collect the body temperature and diaper humidity information of the care recipient through the multi-channel Bluetooth transmission protocol. Figures 4(b) and 4(c) illustrate the workflow chart for smart sensors used to collect physiological data and Bluetooth tags used for indoor positioning, respectively. The Bluetooth hub sends physiological data to the system server. The system server can transmit notifications/alerts (work coordination information exchange or indoor position information) to smartphones via WiFi/4G. Figure 8 shows the system architecture consisting of (a) data from sensors (medical equipment), medical staff (message), and tags (position information), (b) a system server for processing data stored in the database and providing MQTT service, and (c) service modules for medical data monitoring, medical equipment positioning, and message exchanging.
- The presented system, to the best of the authors’ knowledge, for the first time, looks into the function of medical staff work coordination information exchange and applies smart T-shirts and smart diapers to care recipients. Moreover, a secure MQTT is proposed to tackle the security problem of IoT. The presented system has successfully integrated physiological information sensing technology and wearable devices with mobile positioning technology in an Internet of Things-based medical care and medical resource management platform, solving the problem of medical equipment rental management, allowing medical staff and medical management staff to share medical information to improve the efficiency of medical institutions, reducing the workload of nursing and medical management staff and improving the efficiency and quality of care with limited human resources.
- The explanation of relevant equations has been added in the manuscript marked in red color.
- In order to improve the logic of the article, motivations and challenges of this study, discussions about important related work, and the contributions of this paper have been added in the introduction. The logic of the article needs further improvement.
Reviewer 4 Report
Let me point out the very important general task of the paper: “The Internet of Things (IoT) is con- 14 structed for this medical care purpose. The collected data are classified and used to monitor the 15 status of patients in real-time with a Secure MQTT mechanism. The measured physiological signals 16 are also used for developing an IPS.”
It is so important thesis for the general idea of the methodology and step by step research work, leading to the final results and summary.
I also recommend the following paper to be add to the references. The content of this paper is closely related to the main task of the paper by: “KamiÅ„ska Dorota, SmóÅ‚ka Krzysztof, ZwoliÅ„ski Grzegorz, Wiak SÅ‚awomir, Merecz-Kot Dorota, Anbarjafari Gholamreza: Stress Reduction Using Bilateral Stimulation in Virtual Reality. IEEE Access, 2020, vol. 8, no , p. 200351-200366”.
Generally, in my opinion, this paper is high rank scientific knowledge. Telling more with wide range of application in near future. I strongly recommend this paper for publication.
I also recommend to update the text of the paper to reduce simple language mistakes.
Author Response
Many thanks to the reviewer’s comments and encouragement. We have also done our best to update the text of this manuscript to reduce simple language mistakes. The recommended paper has been added in the References and discussed in the Introduction as follows.
“Different types of physiological signals were acquired using GSR (Galvanic Skin Reaction), EMG (Electromyography) and HRM (Heart Rate Monitor) sensors and were analyzed in terms of variability over time before, during, and after the EMDR (Eye Movement Desensitisation and Reprocessing) session [16], in which a VR (Virtual Reality) with the bilateral stimulation used in as a tool to relieve stress was proposed. A 15 minutes relaxation training program was created for adults in a virtual, relaxing environment.”
- KamiÅ„ska, D.; SmóÅ‚ka, K.; ZwoliÅ„ski, G.; Wiak, S.; Merecz-Kot, D.; Anbarjafari, G. A. Stress Reduction Using Bilateral Stimulation in Virtual Reality. IEEE Access, 2020, 8, 200351-200366.
Reviewer 5 Report
The paper contains a clean description of IoT system to monitor patients. For that, authors provided a clear description of architecture system and experimental campaign to check for correct algorithms operation.
The paper is a solid work with good technical contents with a completeness and accuracy of references and figures. The importance and timeliness of the topic addressed in the paper within its area of research is good.
In IoT topic, It is necessary some consideration about communication security. The authors can be insert one paragraph in the introduction section, for example based on this paper:
L. Mucchi, S. Jayousi, A. Martinelli, S. Caputo, and P. Marcocci, “An Overview of Security Threats, Solutions and Challenges in WBANs for Healthcare,” 2019 13th International Symposium on Medical Information and Communication Technology (ISMICT). IEEE, May 2019. doi: 10.1109/ismict.2019.8743798.
Author Response
The following discussion about communication security [17][18] is added in the introduction.
“The data from the sensors placed in or on the patient’s body contain personal privacy, and their transmission out of the body towards a medical server, therefore, requires communication security. Azbeg et al. [17] presented a secure healthcare system that integrates IoT with Blockchain to support remote patient monitoring, especially when it comes to chronic diseases that need regular monitoring. The security was ensured by using the re-encryption proxy together with Blockchain to store hash data via IPFS (Inter Planetary File System). Mucchi et al. [18] reviewed the security challenges and solutions for WBAN (Wireless Body Area Network). The challenges include data confidentiality, data authentication, data integrity, data freshness, secure management, data availability, efficiency and usability, authorization, non-repudiation, scalability up and downsizing, flexibility, data security, and reliability. Some security strategies that satisfy the requirements for securing internal communication in WBAN are avoiding DoS (denial-of-service) attacks, avoiding data tampering attacks, achieving data confidentiality, and avoiding data authenticity. Discretionary Access Control (DAC), Mandatory Access Control, Role-Based Access Control (RBBAC), and Attribute-Based Encryption (ABE) are some of the existing access control adopted algorithms. A trade-off among security, efficiency, flexibility, and usability should be made for new solutions to cope with the future trend of the development of smaller and low power consumption medical sensors.”
- Mucchi, L.; Javousi, S.; Martinelli, A.; Caputo, S.; Marcocci, P. An Overview of Security Threads, Solutions and Challenges in WBAN for Healthcare, In Proceedings of the 13th International Symposium on Medical Information and Communication Technology (ISMICT). May 8-10, Oslo Norway, 2019.
Round 2
Reviewer 2 Report
The authors have successfully responded to comments.
Reviewer 3 Report
The author has made revisions according to the review comments, and I agree to accept the publication of this article.
Reviewer 5 Report
Authors followed the suggestions of the reviewers.